**EMBO** *reports*

# A bacterial toxin-antitoxin system involved in an unusual response to genotoxic stress

Jordan D Lin[1,3], Beth Nicholson [ID]2 & Alexander W Ensminger [ID]1,2✉

## Abstract

**To contend with environmental challenges, bacteria have evolved numerous stress response pathways. A notable example is the adoption of a dormant state called persistence, whereby cells reversibly restrict their growth and await favorable conditions. The genetics of persistence remain poorly understood, and genes called toxin–antitoxin (TA) systems have controversially been implicated in this phenotype. To examine their role in persistence, we construct a pan-TA deletion strain of the bacterial pathogen *Legionella pneumophila* and test its capacity to survive diverse stresses. We identify a single predicted TA system, GndRX, that under genotoxic stress conditions leads to cell death rather than promoting survival, whereas Δ*gndRX* cells adopt a viable but nonculturable state. Strikingly, this enhanced survival is conferred to wild-type cells in a contact-dependent manner during co-culture. Despite having homology to other TA systems, GndRX displays non-canonical activity, and we hypothesize that it has undergone functional domestication by the cell. Overall, our work reveals both a new physiological function for TA systems in bacteria as well as a heretofore undescribed phenomenon of contact-dependent survival within persister cells.**

**Keywords** Persistence; Toxin–antitoxin System; *Legionella pneumophila*; Genotoxic Stress; Cell Death
**Subject Categories** Microbiology, Virology & Host Pathogen Interaction; Signal Transduction

## Introduction

Bacteria are exposed to frequent and diverse cellular stresses throughout their life cycle. Consequently, much of bacterial life is spent in periods of adversity, punctuated by brief transitions to growth-favoring conditions. To survive these challenges, bacteria have evolved a myriad of different stress tolerance strategies. For example, in response to or in preparation for harsh and unpredictable conditions, bacteria transition to various states along a continuum of metabolic restriction and reversible growth arrest that are collectively termed "dormancy" (Lennon and Jones, 2011;

Walker et al, 2024). This behavior can be programmed or stochastic, and importantly, can render cells recalcitrant to treatments such as disinfection and antibiotics (Bergkessel et al, 2016). Because dormancy encompasses distinct phenotypic programs and is implicated in stress survival, the breadth of genetic pathways regulating cellular inactivity are critical to uncover and characterize (McDonald et al, 2024).

One state of quiescence that is of considerable importance is persistence: a phenomenon wherein a subset of a population is able to survive prolonged killing stress after the majority of cells have died off (Balaban et al, 2019). Persistence is distinct from resistance and tolerance in that it is nonheritable, and instead appears to result from the heterogeneity inherent within any cell population that manifests as a distribution of growth and metabolic rates (Balaban et al, 2004; Harms et al, 2016; Ronneau et al, 2021; Bollen et al, 2023). Slow growing cells with reduced metabolic activity—the consequence of a variety of proposed mechanisms (Orman and Brynildsen, 2013; Levin et al, 2014; Pontes and Groisman, 2020; Kaplan et al, 2021; Urbaniec et al, 2022; Zou et al, 2022)—therefore constitute a persister subpopulation that is transiently tolerant of the many stresses to which growing cells are susceptible. Importantly, bacterial persisters have been observed in virtually all studied organisms (Dawson et al, 2011), and owing to the recalcitrance of this subpopulation to drug treatment, are a source of chronic or recurrent infections (Fisher et al, 2017; Gollan et al, 2019) and the development of antibiotic resistance (Cohen et al, 2013; Levin-Reisman et al, 2017; Barrett et al, 2019; Liu et al, 2020).

The genetics of persistence remain complex and unresolved, with numerous genes and pathways proposed thus far (Hu and Coates, 2005; Spoering et al, 2006; Dörr et al, 2010; Ma et al, 2010; Shan et al, 2015; Conlon et al, 2016; Shan et al, 2017; Cameron et al, 2018; Lopatkin et al, 2019; Pu et al, 2019; Wilmaerts et al, 2019; Pacios et al, 2020; Wood and Song, 2020; Personnic et al, 2021; Mohiuddin et al, 2022). Among the strongest candidates are toxin–antitoxin (TA) systems: nearly ubiquitous yet poorly understood genetic modules in bacteria that can function as growth toggling switches (Harms et al, 2018). TA systems have long been implicated in the persister phenotype, particularly given the abundance of chromosomal modules with no ascribed function and their capacity for dormancy-inducing cellular toxicity and detoxification leading to resuscitation (Rotem et al, 2010). However, the involvement of TA systems in persistence has been the subject of substantial controversy and is now strongly debated (Vázquez-Laslop et al, 2006; Ramisetty et al, 2016a; Kim and Wood,

[1]Department of Molecular Genetics, University of Toronto, Toronto, ON, Canada. [2]Department of Biochemistry, University of Toronto, Toronto, ON, Canada. [3]Present address: Department of Medicine, Stanford University, Stanford, CA, USA. ✉E-mail: alex.ensminger@utoronto.ca

2016; Van Melderen and Wood, 2017; Harms et al, 2017; Goormaghtigh et al, 2018; Ronneau and Helaine, 2019; Fraikin et al, 2020; Jurėnas et al, 2022). Arguing against a primary function in persistence is the growing evidence of critical roles for TA systems in bacterial immunity and defense against bacteriophage (Lopatina et al, 2020; LeRoux and Laub, 2022; Kelly et al, 2023; Laub and Typas, 2024). Currently, additional work is required to test the hypothesis of TA system involvement in persistence, preferably using new bacterial systems and with an emphasis on deletion phenotypes rather than overexpression studies (Fraikin et al, 2020). Given their high copy number in many bacterial genomes, only a few attempts to construct pan-TA deletion strains have been undertaken (Conlon et al, 2016; Harms et al, 2017; Goormaghtigh et al, 2018; Pontes and Groisman, 2019; Rosendahl et al, 2020), yet even with these strains the breadth of stresses tested were by no means exhaustive and several systems likely remained (Hossain et al, 2021). As constructing numerous successive deletions increases the risk of unwanted effects on a genome, a bacterium with a minimalist TA repertoire would be an optimal model in which to perform such work.

In its natural environment, the bacterial pathogen *Legionella pneumophila* encounters highly stressful conditions as it transitions between replication within diverse protozoan hosts and an extracellular non-replicative state (Oliva et al, 2018; Garduño, 2020). Once internalized within a host cell, *L. pneumophila* must withstand the inhospitable intracellular environment, in addition to nutrient limitation within the restrictive vacuolar compartment in which it resides (Shames, 2023). Outside of the host, *L. pneumophila* must contend with a large range of environmental conditions, including extremes of temperature and osmolarity (Molofsky and Swanson, 2004). Found in most global freshwater systems, *L. pneumophila* encodes seven predicted TA systems of unknown function (Xie et al, 2018). Given this small number of systems, we sought to leverage this bacterial model to test how a strain devoid of TA systems tolerates diverse extracellular stresses.

Here, we report the construction of a pan-TA deletion strain in *L. pneumophila* (Δ7TA) that serves to address the question of the function of TA systems in bacterial cells. Compared with wild-type *L. pneumophila*, the Δ7TA strain exhibits enhanced cell survival specifically under conditions of genotoxic stress, both at the global population level and within the persister subpopulation. This is the consequence of the deletion of a single predicted TA module (*lpg1604-05*), which encodes a RES-Xre type system that is surprisingly non-toxic to the cell. Thus, contrary to a model in which TA systems increase survival and persistence, the presence of this system leads to rapid cell death during DNA stress, and we show that this is the consequence of the depletion of cellular NAD+. Because of this, we named these genes *gndR* and *gndX*, for genotoxic stress-induced NAD+ depletion leading to cell death, RES and Xre domain-containing proteins, respectively. Interestingly, the enhanced survival response is accompanied by the production of the metabolite homogentisic acid (HGA) and considerable transcriptional divergence between the wild-type and Δ*gndRX* strains, with the bulk Δ*gndRX* population appearing to adopt a dormant cellular state. Strikingly, enhanced persister survival can be conferred from the Δ*gndRX* strain to wild-type cells in a contact-dependent manner, and this is dependent on the ratio of Δ*gndRX* to wild-type cells. Overall, these findings reveal the participation of a non-canonical TA system in a previously undiscovered response to cellular stress in bacteria that is reminiscent

of regulated cell death, and a contact-dependent mechanism of persister survival that is suggestive of complex intercellular communication dynamics within the bacterial population.

# Results

## Construction and validation of the *L. pneumophila* Δ7TA strain

The genome of *L. pneumophila* str. Philadelphia-1 encodes seven predicted TA systems (Fig. 1A)—a comparatively small set relative to other studied bacteria (Xie et al, 2018). These systems are unevenly distributed across the *Legionella* species phylogeny (Fig. 1B); in some instances, they display patchy conservation suggestive of horizontal exchange, whereas in other cases these systems are found in the majority of species and appear to be vertically inherited. Of the TA systems predicted in *L. pneumophila*, all are classified as type II and thus encode both a protein toxin and antitoxin in a putative operon. To leverage this reduced genomic TA repertoire and systematically probe their effect on the cell, we constructed multiple parallel lineages of a pan-TA deletion strain (the Δ7TA strain) using a scar-free recombineering technique. In addition to this, we constructed single TA deletion strains for each of the seven modules. The resulting mutant strains were verified both by Sanger and next-generation sequencing to ensure the fidelity of the edits (Fig. EV1) and to catalog any background genomic mutations. Reference assembly to the *L. pneumophila* genome identified a small number of single-nucleotide polymorphisms (SNPs), which were controlled for across the parallel Δ7TA lineages (Appendix Table S1). In addition, we assembled unmapped sequencing reads and queried them against the NCBI nr database to ensure that no reads were assigned to phage or sources other than common lab contaminants. Finally, we de novo assembled each genome and compared it to the reference genome to confirm the absence of any genomic rearrangements.

## The Δ7TA strain shows enhanced survival and HGA production in response to genotoxic stress

We next compared two Δ7TA lineages with wild-type *L. pneumophila* under a variety of conditions to ascertain whether the absence of these seven systems impacts *L. pneumophila* replication or stress tolerance. We observed no difference between strains during broth growth (Fig. 2A), multiplication within U937-derived monocytes (Fig. 2B), or prolonged stationary phase (Fig. 2C). We then examined whether there was an effect on persistence during treatment with antibiotics targeting three different cellular processes. Disruption of protein (Fig. 2D) and cell wall (Fig. 2E) synthesis did not yield differing survival kinetics across strains, however, exposure to DNA stress via the fluoroquinolone ciprofloxacin produced a substantial divergence in persister survival (Fig. 2F). Surprisingly, the Δ7TA lineages showed increased survival relative to the wild-type strain under these conditions and these survival differences were only observed after prolonged stress exposure when the populations had already undergone the biphasic killing kinetics that are a hallmark of persistence (Fig. EV2A). This indicated that the phenomenon occurs within the persister subpopulation and is not the

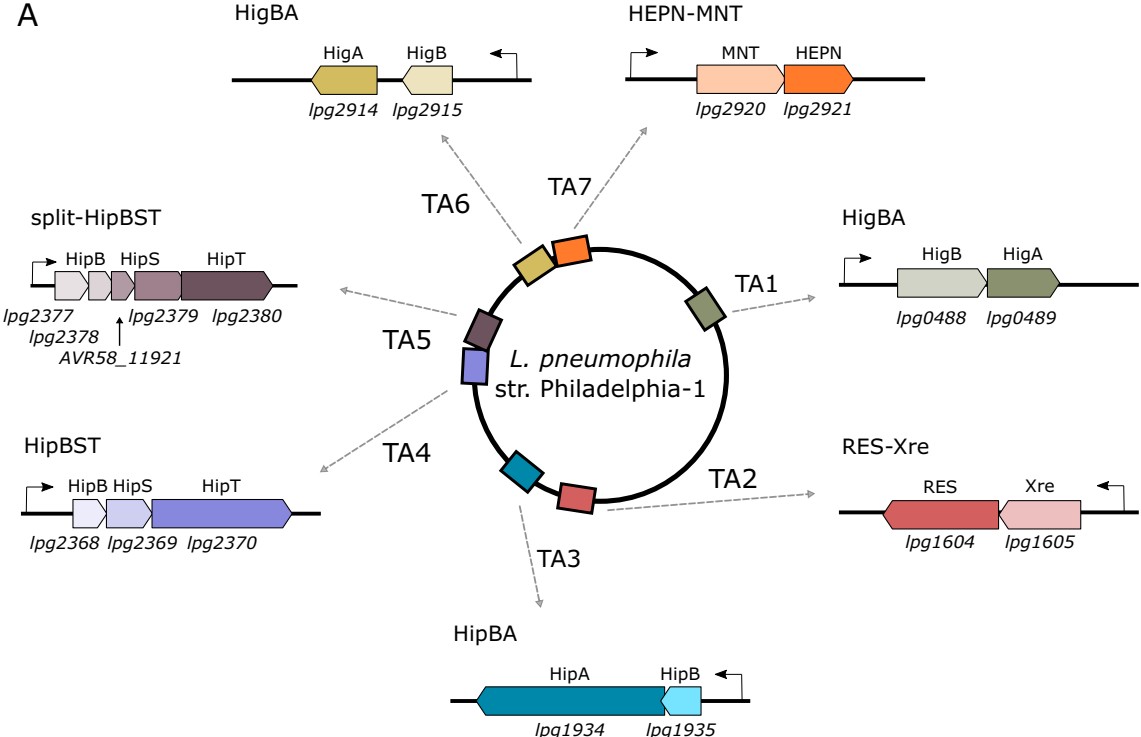

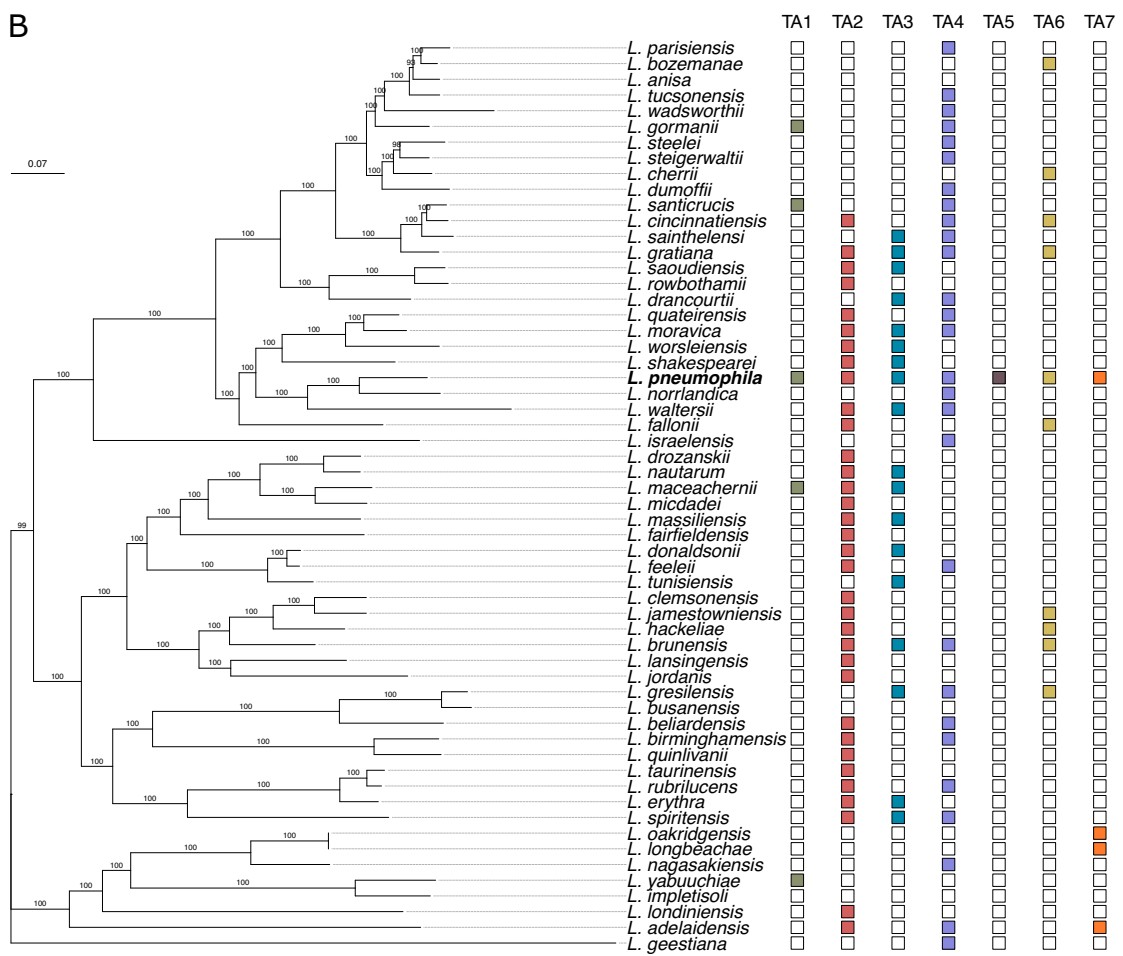

**Figure 1. Construction of a pan-TA deletion strain in *L. pneumophila*.**

(A) Landscape of predicted toxin–antitoxin systems in the genome of *Legionella pneumophila* str. Philadelphia-1 (Xie et al, 2018). (B) Core genome phylogeny of 58 species in the *Legionella* genus displaying the conservation of seven predicted TA systems from the *L. pneumophila* genome. The phylogeny was constructed as described previously (Lin et al, 2023). The scale bar denotes substitutions per site, and bootstrap values are provided for each node.

consequence of increased stress tolerance within the entire population. Consistent with this, we confirmed that the minimum inhibitory concentration (MIC) of ciprofloxacin was the same across strains (Fig. EV2B) and that our treatment concentration (25 μg/mL) far exceeded the MIC (>100×). Furthermore, we did not observe any differences in antibiotic susceptibility or resistance between strains (Fig. EV2C,D).

Coincident with enhanced persister survival, we also observed the production of brown pigment in the Δ7TA cultures after 24 h of genotoxic stress (Fig. 2G). *L. pneumophila* is known to synthesize and secrete the metabolite homogentisic acid (HGA) (Steinert et al, 1995), which subsequently becomes oxidized and polymerizes outside of the cell, resulting in brown pigmentation. HGA secretion by *L. pneumophila* typically occurs during the stationary phase and can be detected by measuring absorbance at 400 nm (Levin et al, 2019). We therefore quantified HGA production and observed increased secretion by the Δ7TA strain after exposure to ciprofloxacin, suggesting that HGA biosynthesis or secretion is influenced by the activity of one or multiple TA systems. Interestingly, treatment with the antibiotics carbenicillin or gentamicin had no effect on HGA production (Fig. EV2E), suggesting a link between persister survival and metabolite biosynthesis that is specific to the response to DNA stress.

To further confirm that this was a persistence phenotype, rather than being the consequence of altered tolerance or resistance, we repeated the ciprofloxacin killing assay with colonies recovered after an initial round of drug treatment. This second generation of cells survived equivalently to the prior generation (Fig. 2H), indicating that no heritable changes in drug sensitivity were present in these cell populations. Next, we tested an alternative mode of producing genotoxic stress to determine whether the observed response was specific to fluoroquinolone antibiotics or more general DNA damage. Treatment with the chemotherapeutic mitomycin C rapidly eradicated persister cells (Fig. EV2F), consistent with its potential in treating persistent infections (Kwan et al, 2015), yet HGA production differences were comparable to those observed with ciprofloxacin. As an alternative to time-kill assays, which use high drug concentrations, we next performed dose–response growth curves with mitomycin C at concentrations above and below the MIC (Fig. EV2G). From this, we observed an increase in the concentration of drug required to inhibit the growth of the Δ7TA strain relative to the wild-type (Fig. EV2H) and enhanced survival of the Δ7TA strain during prolonged exposure to doses at or above the MIC (Fig. EV2I). This was therefore distinct from the persister phenotype observed with ciprofloxacin and indicative of both subtle susceptibility differences between strains and a DNA stress-related response that is not specific to fluoroquinolones. In summary, the deletion of all predicted TA systems in the *L. pneumophila* genome resulted in enhanced— rather than diminished—persister survival to antibiotic killing when the mode of action is damage to cellular DNA. This was accompanied by the production of the cryptic metabolite HGA and

not the consequence of heritable changes in drug sensitivity. Interestingly, treatment with an alternative DNA-damaging agent, mitomycin C, replicated the HGA secretion results and revealed dose-dependent differences in susceptibility between strains at the population level.

## The *gndRX* locus is responsible for the genotoxic stress response phenotype

Rather than proceeding further with the Δ7TA strain, we instead investigated whether one or multiple TA systems are involved in the response to genotoxic stress. To this end, we tested each individual TA deletion strain for enhanced survival and HGA production during ciprofloxacin-induced DNA damage. From this, we found that only one putative TA system deletion (ΔTA2), comprising the *gndRX* (*lpg1604-05*) locus, phenocopied the Δ7TA strain for this response (Fig. 3A). We validated the involvement of this system with genomic complementation of the wild-type *gndRX* locus in both the Δ*gndRX* (Fig. 3B) and Δ7TA (Fig. EV3A) deletion backgrounds. In both complemented strains, wild-type survival and HGA production kinetics were restored. Furthermore, we compared survival between the wild-type and Δ*gndRX* strains with an additional genotoxic stressor, ultraviolet (UV) radiation. From this, we observed similar survival dynamics to those of mitomycin C treatment, with increasing doses of UV reducing growth of the wild-type relative to the Δ*gndRX* strain (Fig. EV3B). These results therefore confirmed that the *gndRX* locus alone is responsible for the genotoxic stress response observed with the Δ7TA strain and that the presence of this system influences the cell's susceptibility to DNA stress.

Within the *L. pneumophila* TA repertoire, GndRX is the most conserved system across the *Legionella* species phylogeny (Fig. 1B) and is found in more than half of the genomes surveyed (34/58). We compared homologs of both proteins across *Legionella* species and found that the official annotation of GndX (Kanehisa and Goto, 2000; Sayers et al, 2021; UniProt Consortium, 2023) appears to contain an additional 35 amino acids at the N-terminus that is not conserved in any other homolog sequence (Fig. EV3C) and thus is likely a misannotation. As we had constructed our deletions based on this annotation, and to ensure that the inclusion of the additional sequence did not account for our observed phenotypes, we restored this misannotated sequence in the Δ7TA strain. Subsequent genotoxic survival and HGA production assays demonstrated that the fixed Δ*gndRX* deletion in the Δ7TA background phenocopied the original Δ7TA strain (Fig. EV3D). Consistent with this, we also observed a restoration of wild-type death kinetics when the original Δ*gndRX* strain (missing the 105 bp) was complemented in trans using a plasmid encoding the correct *gndRX* sequence without the additional 105 bp upstream (Fig. EV3E). These findings further confirmed the involvement of the *gndRX* system and demonstrated that the misannotated upstream region did not account for the response to genotoxic

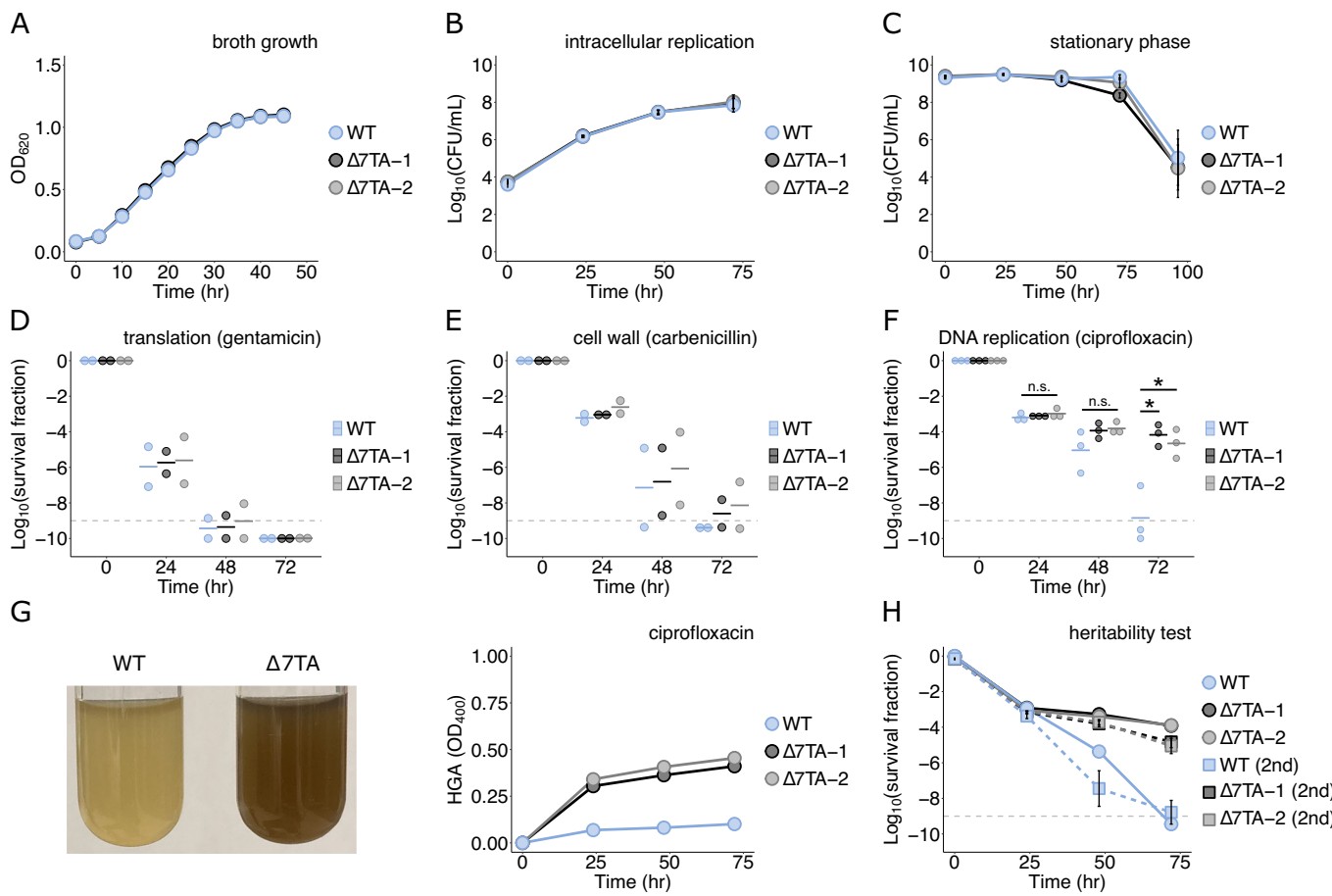

**Figure 2. The *L. pneumophila* Δ7TA strain shows enhanced survival following genotoxic stress.**

Comparisons between wild-type (WT) and two *L. pneumophila* Δ7TA lineages for (**A**) growth in rich broth (data are representative of $n = 3$ biological replicates), (**B**) replication within differentiated U937 monocytes ($n = 3$ biological replicates), or (**C**) survival during prolonged stationary phase ($n = 3$ biological replicates). Time-kill assays measuring survival during treatment with (**D**) gentamicin ($n = 2$ biological replicates), (**E**) carbenicillin ($n = 2$ two biological replicates), and (**F**) ciprofloxacin ($n = 3$ biological replicates). (**G**) Homogentisic acid (HGA) production after 24 h treatment with ciprofloxacin (left) and quantification over 72 h genotoxic stress (right) are shown from a representative experiment ($n = 3$ biological replicates). (**H**) Time-kill assay measuring the heritability of drug tolerance after exposure to ciprofloxacin. In the first generation (circles and solid lines), the WT and Δ7TA strains were treated with ciprofloxacin as above, and survival was quantified. In the second generation (squares and dashed lines), $n = 3$ biological replicates taken from surviving colonies from each strain in the first experiment at the 24 h timepoint were again treated with ciprofloxacin and survival was quantified. Data information: In (**B**, **C**, **H**), data are the mean (averaged for clarity) ± SEM. (**D–F**) The bar represents the mean. (**F**) Statistical hypothesis testing was performed with the Welch's *t* test (n.s. = not significant; *$P < 0.05$). The limit of detection on all applicable plots is indicated with a dashed gray line. Source data are available online for this figure.

stress. Since we had already whole-genome sequenced and worked extensively with the original Δ*gndRX* strain, we chose to continue experiments in this genetic background. Finally, and notably, the GndRX system is also the most conserved TA system within *L. pneumophila* strain genomes (Fig. EV3F), where it is present in all complete genomes in the NCBI Refseq database ($n = 117$; Dataset EV1). This system therefore constitutes part of the *L. pneumophila* core genome, which is an unusually high level of conservation for a TA module.

## GndRX is a predicted RES-Xre TA system, but does not possess canonical functionality

The *gndRX* locus encodes a predicted RES-Xre type TA system, with GndR containing the toxin-associated RES domain and GndX containing both a DNA-binding helix-turn-helix domain and an

Xre/MbcA/ParS toxin-binding domain (Fig. 4A). The RES domain is comprised of three conserved residues (arginine, glutamate, serine) that act as a putative catalytic triad within a central pocket. To begin characterizing the GndRX system, we first sought to identify related proteins using the homology search tools HHpred (Söding et al, 2005) and Foldseek (van Kempen et al, 2023). From this, we identified only five high confidence hits in the Protein Data Bank, and these were all to other RES-Xre TA systems that have been recently reported: NatRT (Santi et al, 2024), MbcTA (Freire et al, 2019), VPA0770-0769 (Zhang et al, 2023), ParST (Piscotta et al, 2019), and RES-Xre_Pp (Skjerning et al, 2019) (Figs. 4B,C and EV4A). The toxins from these systems bear only remote sequence homology to GndR (Fig. EV4B) but all possess the conserved R-E-S residues (Fig. 4C) and their RES domain folds (Fig. EV4A) are highly similar to the AlphaFold (Jumper et al, 2021; Varadi et al, 2024) predicted structure of GndR (Fig. 4D). In particular, the

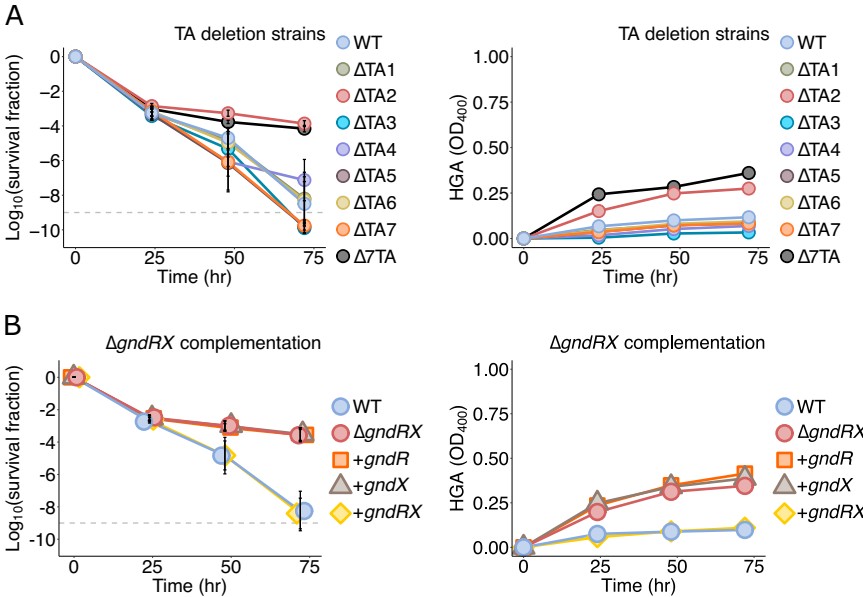

**Figure 3. The *gndRX* locus is responsible for the genotoxic stress response phenotype.**

Time-kill assays (left) and HGA measurements (right) showing treatment with ciprofloxacin for (**A**) wild-type, single ΔTA, and Δ7TA-1 *L. pneumophila* strains, and (**B**) chromosomal complementation strains of the *gndRX* system in the Δ*gndRX* background. For each strain in (**B**), the chromosomal Δ*gndRX* lesion was repaired with a linear PCR product encoding the desired genetic sequence and confirmed with Sanger sequencing. Survival data from $n = 3$ biological replicates and HGA production measurements from representative experiments ($n = 3$ biological replicates) are shown. Data information: In (**A, B**), survival data are presented as mean (averaged for clarity) ± SEM. The limit of detection on all applicable plots is indicated with a dashed gray line. Source data are available online for this figure.

recently reported NatT toxin from *Pseudomonas aeruginosa* is the most closely related homolog (Fig. EV4B) and contains an N-terminal "flap" motif (Fig. EV4A) that is both absent from the other TA systems and predicted in the GndR structure (Fig. 4D). As these five RES-Xre homologs have been previously characterized, this offered the potential to compare the functional capacity of GndRX across all members of this family of TA systems.

Despite their shared RES domain, the homolog toxins perform different catalytic activities: NatT and MbcT are $NAD^+$ phosphorylases, VPA0770 is a predicted ADP-ribosyltransferase (ART), ParT is an ART, and $RES^{Pp}$ is a predicted $NAD^+$ glycohydrolase. Thus, these proteins all consume or modify $NAD^+$ to arrest cellular growth and share a similar $NAD^+$ binding pocket in which the R-E-S residues coordinate this substrate. To test the importance of the R-E-S residues in the genotoxic stress response phenotype, we substituted each site with alanine at the chromosomal locus in the wild-type background and performed time-kill assays with ciprofloxacin. These substitutions increased survival and HGA production during genotoxic stress (Figs. 4E and EV4C), thereby phenocopying the response of the Δ*gndRX* strain and suggesting that GndR activity is necessary for the functioning of the system. Surprisingly, genomic complementation of either *gndR* or *gndX* alone did not result in a wild-type response to stress (Fig. 3B,C), suggesting that both proteins are required for normal functionality rather than just GndR. These findings were reminiscent of the NatRT system, where the NatR antitoxin is required for NatT toxin functionality (Santi et al, 2024). Consequently, NatT is not toxic alone and instead requires NatR to arrest growth, albeit at a low relative abundance. To test whether this property is similar for GndRX, we ectopically expressed each protein individually and in

combination in *L. pneumophila* Δ*gndRX* cells (Fig. 4F). Surprisingly, no growth inhibition was observed with overexpression of either protein alone, and only a weak effect was present at the highest levels of GndR expression relative to GndX. In addition, while NatT is able to arrest growth in *E. coli* cells (Santi et al, 2024), we did not observed a similar behavior for GndR (Fig. EV4D). These findings suggest that while GndRX is predicted to be a TA system, it does not appear to function in a canonical manner and instead has diverged from other RES-Xre TA homologs.

The lack of toxicity for the GndRX system led us to wonder whether it retains other characteristics of type II TA systems. One such property is the physical interaction between toxin and antitoxin proteins to facilitate system regulation and neutralization. We tested whether direct binding between GndR and GndX can occur using the yeast two-hybrid (Y2H) assay, and observed that both proteins can stably associate (Fig. EV4E). To model this interaction, we predicted the multimeric structure of GndR-GndX using AlphaFold2 (Fig. EV4F). In this model, a C-terminal alpha helix in GndX is predicted to impinge upon the catalytic pocket of GndR where $NAD^+$ would be bound, which is a mode of neutralization observed in other RES-Xre systems, and the physical interaction between proteins is largely facilitated by the ~45 amino acid motif in the N-terminus of GndR (Figs. 4D and EV4F). This same conformation was observed for the N-terminal 'flap' motif in the NatRT system, with the authors reporting an activating mutation (NatT$_{E29D}$) that regulates the hinge of the flap and consequently access for the neutralizing NatR helix into the NatT catalytic pocket (Santi et al, 2024). Interestingly, the E29 residue is conserved in the alignment between NatT and GndR (E32) and is also found in the N-terminal motif (Fig. EV4F). As NatT$_{E29D}$

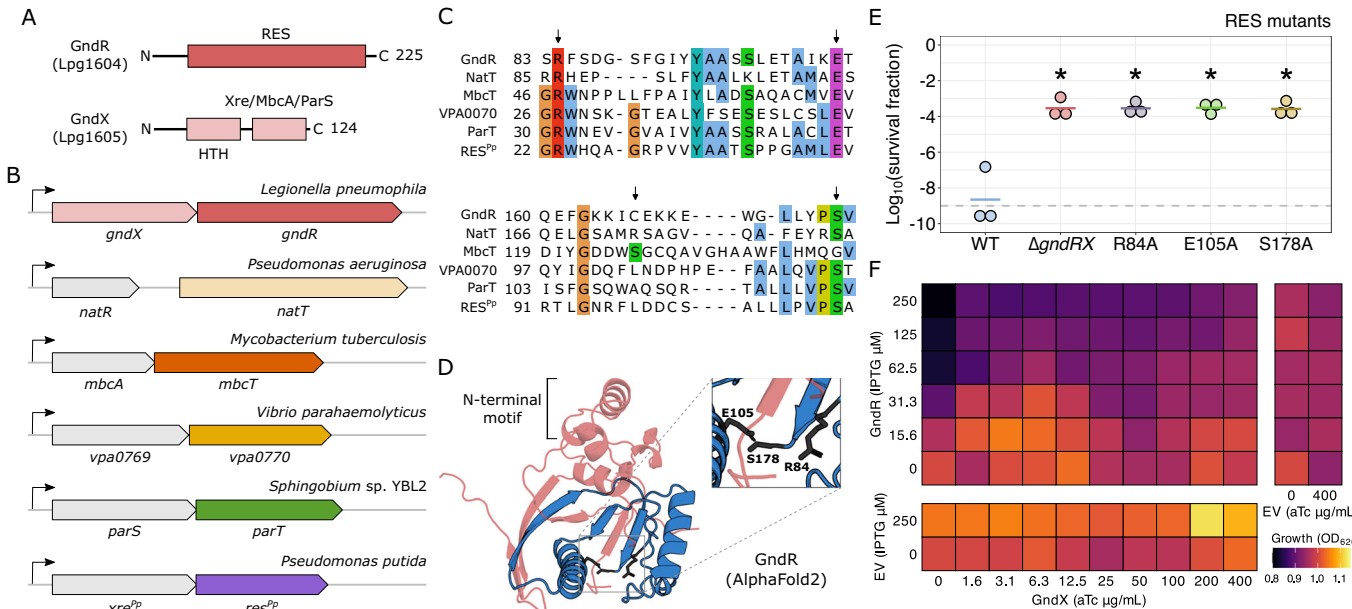

**Figure 4. GndRX has remote homology to RES-Xre TA systems but non-canonical activity.**

(A) Schematic of the domains predicted in the GndR and GndX proteins. (B) Schematic of five bacterial TA systems that have remote homology to GndRX. These were the only protein homologs—TA systems or otherwise—detected for this system. (C) Multiple sequence alignment (MUSCLE) of GndR with the NatT, MbcT, ParT, VPA0770, and RES[Pp] toxins. The R-E-S residues in each sequence are indicated with arrows. Residues conserved in at least 50% of proteins are colored with the Clustal X color scheme as implemented in Jalview (Waterhouse et al, 2009). (D) AlphaFold2 predicted structure of GndR (AF-Q5ZV38-F1-v4) from the AlphaFold Structure Database (Varadi et al, 2024). The RES domain is colored in blue, with the R-E-S residues highlighted in the inset. The N-terminal motif, which is absent in the other toxins, is indicated. (E) Time-kill assay with *L. pneumophila* strains containing substitutions of individual R-E-S residues to alanine and treated with ciprofloxacin. Survival after 72 h is shown (*n* = 3 biological replicates). (F) Growth curve data (maximum measured growth) of *L. pneumophila* Δ*gndRX* expressing *gndR* (pNT562), *gndX* (pJB1806), or both genes in combination (*n* = 2 biological replicates). Expression was induced with the indicated concentrations of IPTG and aTc. Data information: In (E), the bar represents the mean and statistical hypothesis testing was performed with the Welch's *t* test by comparing each mutant against the wild-type (**P* < 0.05). (F) Data are presented as the mean for clarity. The limit of detection on all applicable plots is indicated with a dashed gray line. Source data are available online for this figure.

showed enhanced toxicity in *P. aeruginosa* cells, we constructed the equivalent mutation in GndR (E32D) and measured *L. pneumophila* growth during ectopic expression, however no effect was observed (Fig. EV4G). To examine this mutation under the native stoichiometry of the GndRX system, we constructed this mutation on the *L. pneumophila* chromosome (*gndR*[E32D]) and tested this strain for survival during genotoxic stress. Once more however, we observed no effect on system function (Fig. EV4H).

As GndR appears to diverge both structurally and functionally from other RES toxins, we wondered whether homologs of this protein are found across the tree of life. To address this, we searched the UniprotKB database for homologous proteins and detected 3995 related sequences (Dataset EV2). These are present across numerous and diverse bacterial phyla (Fig. EV4I), though there is an enrichment of homologs within the Pseudomonadota phylum and in particular the Gammaproteobacteria and Alpha-proteobacteria. Interestingly, we also identified two homologous proteins in eukaryotic taxa (A0A444BWB2, A0A812QAU7) that are each predicted to contain both RES and Xre domains linked together as a single protein. The existence of such fusion products, if validated, could thus help to explain the requirement for both proteins in system functioning. In summary, GndRX has remote homology solely to RES-Xre TA systems, yet it displays non-canonical TA activity, suggesting that this system may have evolutionarily diverged from an ancestral TA system.

## The genotoxic survival phenotype produces distinct transcriptomic changes and is HGA-independent

Without a clear understanding of the biochemical activity of GndRX, we next investigated the global effect of system activity on the cell during stress by performing comparative transcriptomic analyses of wild-type and Δ*gndRX L. pneumophila* strains prior to and during ciprofloxacin treatment (Fig. 5A). In the absence of stress, the transcriptomes of both strains were nearly identical (Dataset EV3; 24 differentially enriched genes), suggesting that either system expression or function is restricted to conditions of DNA damage in the cell. After 1 h, both strains underwent dramatic transcriptional shifts with the differential enrichment of ~1900 genes within each transcriptome. These shifts were largely congruent, and the number of differentially enriched gene transcripts detected between strains was comparable to the unstressed transcriptomes (Dataset EV3). There were, however, five genes that were differently enriched in either one strain or the other but not both (Fig. 5B). These included *phhA*, the upstream gene in the HGA biosynthetic pathway, which under DNA stress conditions becomes negatively enriched in the wild-type. This is not observed in the Δ*gndRX* strain and thus may explain the disparity in HGA secretion. We also detected a substantial positive enrichment of *gndRX* transcripts in the wild-type after stress induction. In particular, *gndX* (the upstream gene in the pair) is the

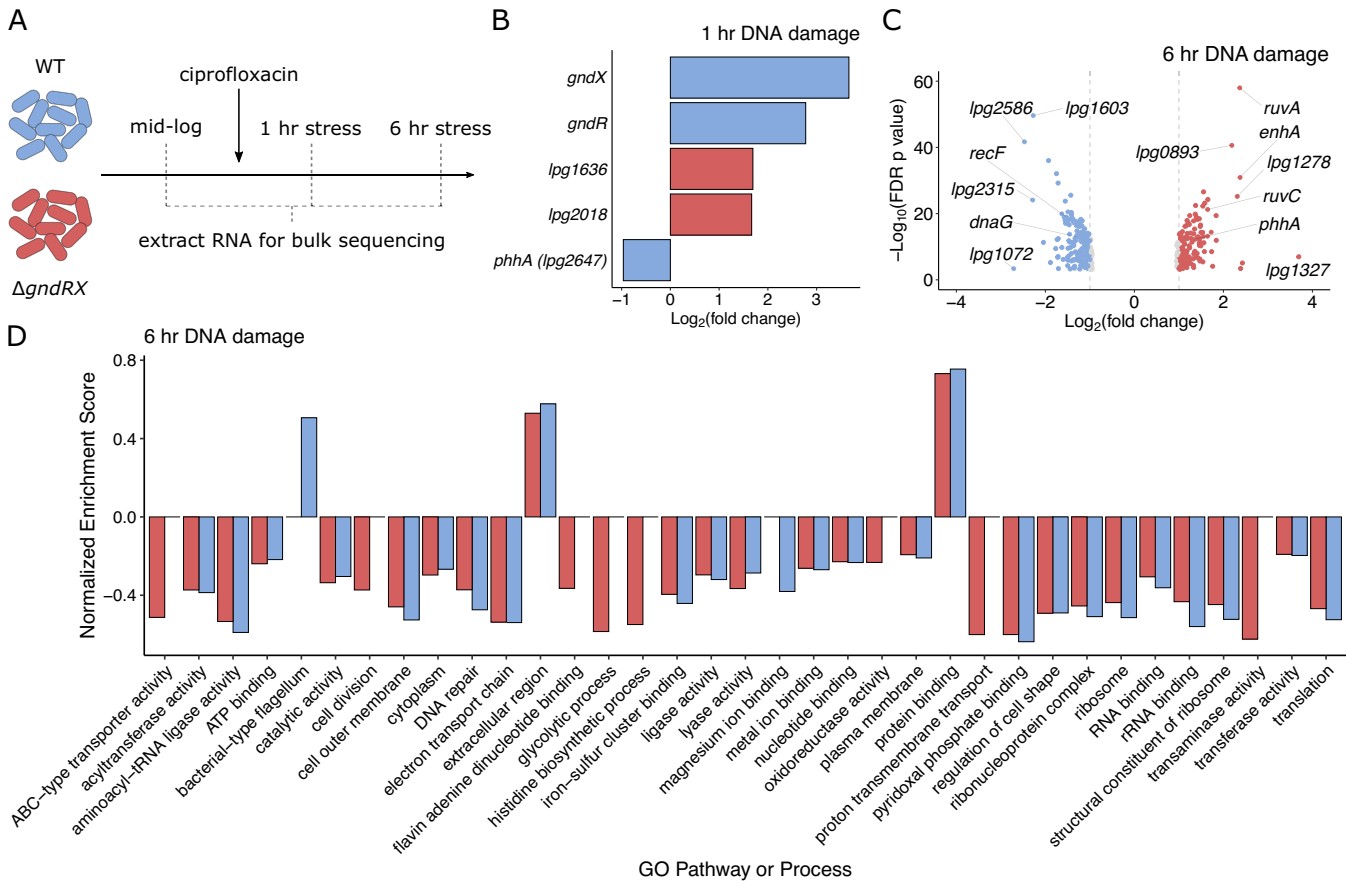

**Figure 5. Genotoxic stress produces divergent transcriptomic responses across strains.**

(A) Overview of the RNA-seq experiment. Wild-type and ΔgndRX strains of *L. pneumophila* were grown to the exponential phase and then treated with ciprofloxacin. At the indicated time points, RNA was extracted and sequenced. (B) Uniquely differentially enriched transcripts in the wild-type and ΔgndRX strains after one hr of ciprofloxacin treatment. (C) Volcano plot of differentially enriched transcripts after six hr of ciprofloxacin-induced genotoxic stress. Positive fold change (x-axis) values indicate enrichment in ΔgndRX relative to the wild-type; negative fold change indicates the opposite. Genes critical in DNA replication and repair (*recF*, *dnaG*, *ruvAC*), HGA biosynthesis (*phhA*), or that are highly differentially enriched across strains are indicated. (D) Bar chart showing gene ontology (GO) term enrichment across both strains after six hr of genotoxic stress. Data are presented in Dataset EV3.

135th most positively enriched transcript we detected, indicating that this system is strongly activated in response to DNA damage.

After 6 h of genotoxic stress, the transcriptomes of both strains diverged substantially (Fig. 5C), with 342 genes identified as differentially enriched (Dataset EV3). Notable among these were the Holliday junction resolvases *ruvA* and *ruvC* (positively enriched in the ΔgndRX strain), the DNA primase *dnaG* (negatively enriched in the ΔgndRX strain), and the recombination repair gene *recF* (positively enriched in the wild-type). Many of the differentially enriched genes are uncharacterized however, such as *lpg1327*, which showed the largest fold change during genotoxic stress but has no predicted function. Interestingly, the gene upstream of *gndRX* (*lpg1603*) was also highly enriched in the wild-type relative to the deletion strain, suggesting a possible transcriptional coupling between these genes. The protein product Lpg1603 is a predicted phospholipase but is otherwise uncharacterized, thus it remains unclear whether this protein has any functional relationship with GndRX. To look at global gene expression patterns, we performed gene set enrichment analysis for gene ontology (GO) annotations (Fig. 5D). This revealed a largely concordant downregulation of many pathways and processes across

strains, however several gene sets were negatively enriched only in the ΔgndRX strain, including those associated with the cell cycle, glycolysis, oxidoreductase activity, and proton transmembrane transport. Conversely, the wild-type showed a positive enrichment of flagellar genes and a negative enrichment of genes involved with magnesium ion binding.

Given the observed differential enrichment of *phhA* between strains, and the reported growth inhibitory properties of HGA (Levin et al, 2019), we hypothesized that the presence of HGA in the culture medium is protective for the ΔgndRX strain during genotoxic stress. To test this, we deleted *phhA* in both the wild-type and ΔgndRX strains and measured their survival compared with their *phhA*+ progenitors during genotoxic stress. Surprisingly, the abrogation of HGA biosynthesis had no impact on the stress tolerance kinetics in either strain (Fig. EV5A), thereby ruling out a role for this molecule in the observed phenotypic response. Though its function in *L. pneumophila* physiology remains uncertain, it has been previously established that HGA secretion primarily occurs in the late exponential and stationary growth phases (Levin et al, 2019). Given this association and its abundant production by the

Δ*gndRX* strain, we wondered whether an alternative growth phase-related phenomenon might influence cell survival, with HGA being indirectly linked. Consistent with this, we observed differences in the magnitude of the survival response during genotoxic stress exposure that were associated with the culture growth phase. Specifically, both strains showed reduced survival in the late exponential relative to the early exponential growth phase, and the survival differences between strains were more pronounced in the late exponential phase (Fig. EV5B). Conversely, this effect was entirely absent when stationary phase cultures were tested. However, these differences were not accompanied by any detectable divergence in the respective transcriptomic environments across growth phases in the absence of stress (Fig. EV5C). In summary, the wild-type and Δ*gndRX* strains undergo distinct transcriptomic changes under genotoxic stress conditions, including the differential enrichment of numerous DNA replication and repair genes and the HGA biosynthetic gene *phhA*. However, despite its abundant secretion by the Δ*gndRX* strain during stress, HGA does not appear to be involved in the cell's survival response, though there is an effect of culture growth phase on persister survival kinetics.

## Enhanced survival is cell-extrinsic and conferrable through a contact-dependent mechanism

Due to the presence of HGA in the culture medium and the growth phase-dependent differences in survival between strains, we wondered if a different metabolite or extracellular factor could facilitate enhanced survival during genotoxic stress. To test this, we conducted co-culture survival assays with the wild-type and Δ*gndRX* strains in which one strain was marked with a luminescent genomic cassette (*lux*) to allow both strains to be distinguished (Fig. 6A). Strikingly, we observed robust and enhanced survival of the wild-type persisters during co-culture with the Δ*gndRX* strain that was comparable to that of the Δ*gndRX* strain by itself (Fig. 6B). Conversely, the presence of the *lux* cassette itself had no impact on the phenotypic response to stress when co-culture was performed in combination with an otherwise identical genetic background. Given these surprising results, we next sought to establish whether the enhanced persister survival conferred to the wild-type strain required cell–cell contact or instead was the consequence of some secreted component of the media. To address this, we repeated the luminescent co-culture experiments in transwell plates containing a 0.1-μm membrane that split each well into two compartments. This design therefore allowed for free diffusion of the culture medium between compartments but prohibited cell–cell contact (Fig. 6C). When physical contact was prevented, this entirely abolished the enhanced survival phenotype conferred to the wild-type cells (Fig. 6C), and this phenotype could be restored by permeabilizing the membrane to allow for cell mixing (Appendix Fig. S1A).

To ensure that the pore size of the membrane was not prohibiting the passage of large extracellular material, such as outer membrane vesicles, we repeated these experiments using a 0.4 μm membrane. Despite the larger pore size, we still observed the same inhibition of conferred survival for the wild-type strain (Fig. 6C). In addition, we validated the permeability of the membranes by measuring HGA diffusion between compartments at the end of each experiment. This demonstrated a net transfer of HGA into the wild-type compartment from the Δ*gndRX* strain

when both were co-cultured that was absent when both compartments contained only wild-type cells (Appendix Fig. S1B). Next, we orthogonally tested for an effect of the culture medium alone by performing survival experiments in which conditioned media from a mixed strain co-culture was removed after 24 h of ciprofloxacin treatment and transplanted into wild-type or Δ*gndRX* cultures that had also been stressed for 24 h. We chose to use conditioned media from mixed co-cultures, rather than from the Δ*gndRX* strain alone, to ensure that any extracellular material that might only be produced in the presence of both strains would be captured. The conditioned media transplant did not affect the survival kinetics of either strain, however, providing further evidence to support that cell–cell contact is required to facilitate enhanced survival of wild-type cells (Fig. 6D). Finally, we sought to test what effect varying the ratio of wild-type to Δ*gndRX* cells would have on survival during co-culture. Unexpectedly, when this ratio was changed to 9:1 (wild-type:Δ*gndRX*), we observed no enhanced survival in either the wild-type or Δ*gndRX* populations (Fig. 6E), suggesting that the overall proportion of Δ*gndRX* cells is important for conferring enhanced persister survival both to the wild-type and within its own population. Overall, these findings demonstrate that *L. pneumophila* Δ*gndRX* cells can confer enhanced survival during genotoxic stress to wild-type *L. pneumophila* in a contact-dependent manner. Furthermore, this enhanced survival is not only cell-extrinsic for wild-type cells, but the survival of Δ*gndRX* cells is also dependent on the density of this population.

## GndRX appears to direct cells to a state of death rather than dormancy during DNA stress

The persister survival differences that we observed between wild-type and Δ*gndRX* strains occur only after a prolonged duration of genotoxic stress, suggesting that other events in the bulk population precede the divergent fates of the persister subpopulations. To investigate this, we measured changes in culture turbidity for both strains during DNA stress. Surprisingly, the two strains experienced highly different death kinetics at the population level during the first 24 h of genotoxic stress exposure (Fig. 7A), despite the number of persisters remaining equivalent during this same period of time (Fig. 7B). Indeed, shortly after stress induction the wild-type strain declined dramatically in culture density, consistent with cell death and lysis within the population. Conversely, the Δ*gndRX* strain maintained a relatively stable culture density, and in fact appeared to increase slightly, possibly indicating ongoing attempts at replication within this population. After 24 h of genotoxic stress, the wild-type population had declined considerably in cell density, indicating a large proportion of cells had died off. The Δ*gndRX* strain, in comparison, retained a similar turbidity to the time zero culture. Interestingly, Δ*gndRX* culture density also began to decline after 24 h (Appendix Fig. S2A), though at a slower rate than the wild-type population. To confirm the presence of cell lysis in the cultures, we utilized a LacZ reporter assay where intracellular LacZ released upon cell lysis into the culture medium can be detected and quantified. We initially detected an enrichment of extracellular LacZ in wild-type cultures relative to the Δ*gndRX* strain over the first 12 h of genotoxic stress (Appendix Fig. S2B). Subsequently, LacZ detection in the Δ*gndRX* culture surpassed that of the wild-type, suggesting increased cell lysis and consistent with the culture turbidity measurements. Notably, cell lysis was detected in the

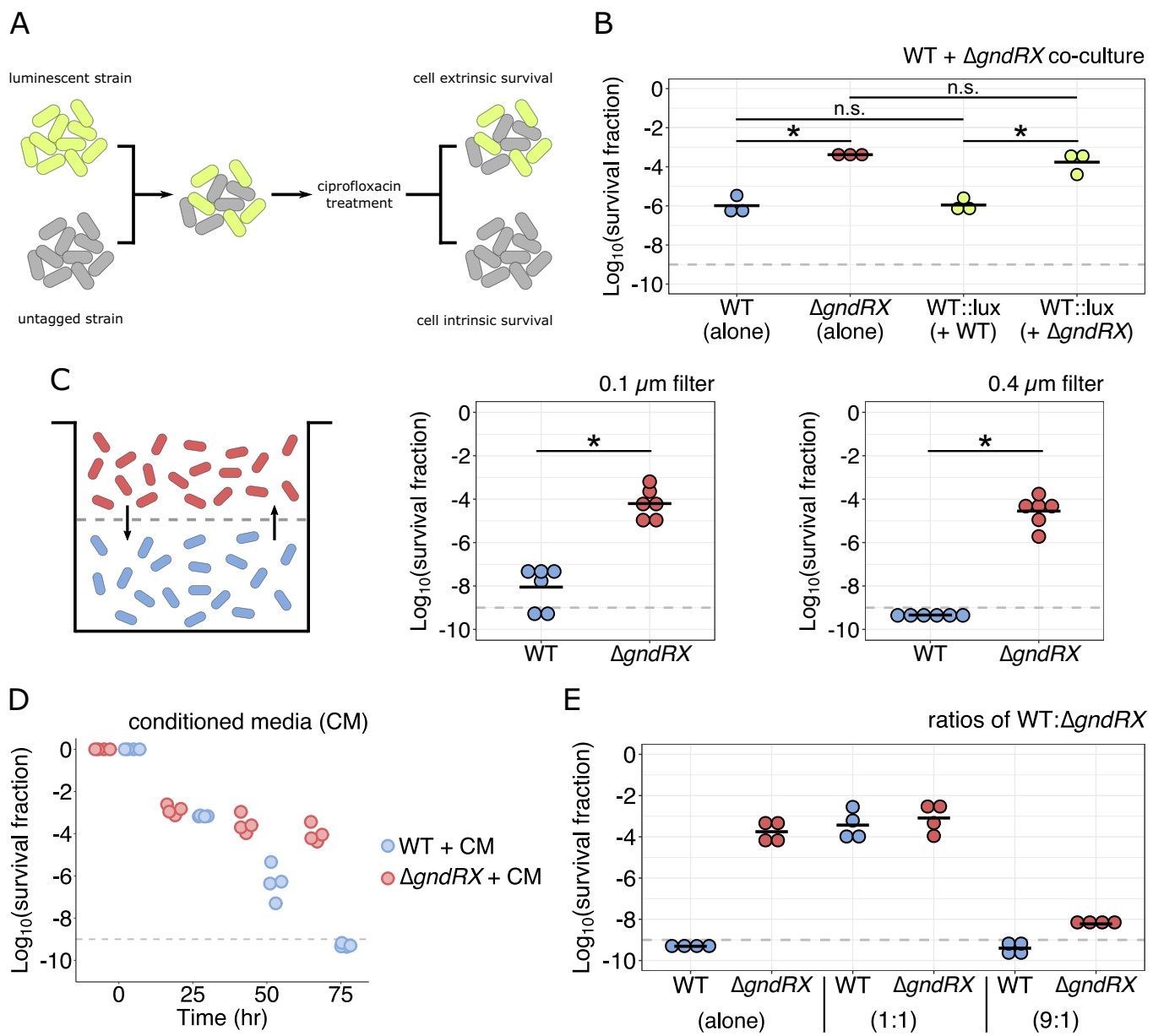

**Figure 6. Enhanced survival is cell-extrinsic and conferred through cell–cell contact.**

(A) Overview of the luminescent co-culture survival experiment. One strain is marked with a luminescent (*lux*) cassette while the other is not. Both strains are mixed and treated with ciprofloxacin for 72 h. The resulting populations can be distinguished via luminescence to determine whether only one strain survives (cell intrinsic survival) or both strains survive equally (cell-extrinsic survival). (B) Ciprofloxacin time-kill assay comparing wild-type *L. pneumophila* containing a chromosomal *lux* cassette co-cultured with either wild-type or Δ*gndRX* cells (*n* = 3 biological replicates). Survival at the 72 h timepoint is shown, and each strain alone is included as a control. (C) Schematic (left) and time-kill assays for survival experiments using transwell plate inserts that prevent cell–cell contact. Wild-type and Δ*gndRX* strains were added to separate compartments of a transwell plate where they were separated by a membrane with the indicated pore size. Cells were then treated with ciprofloxacin for 48 h, and net survival was quantified. Data are from *n* = 3 biological replicates performed in duplicate (each strain with or without the *lux* cassette; i.e., WT:*lux* + Δ*gndRX* and WT + Δ*gndRX*:*lux*). (D) Time-kill assay comparing wild-type and Δ*gndRX* cells that were incubated with conditioned media (CM) from a mixed co-culture of both strains. All cultures (single and mixed) were treated with ciprofloxacin for 24 h, at which point all supernatants were removed and the co-culture conditioned media were transplanted into the individual strains. These cultures were then incubated with the transplanted media for the remainder of the experiment. Data are from *n* = 2 biological replicates performed in duplicate (each strain with or without the *lux* cassette; i.e., WT:*lux* + Δ*gndRX* and WT + Δ*gndRX*:*lux*). (E) Time-kill assays testing differing ratios of WT:Δ*gndRX* cells. Survival of the wild-type and Δ*gndRX* cultures alone is shown on the left. Survival of the wild-type and Δ*gndRX* cultures when mixed 1:1 is shown on the middle. Survival of the wild-type and Δ*gndRX* cultures when mixed 9:1 is shown on the right. Data are from *n* = 2 biological replicates performed in duplicate (each strain with or without the *lux* cassette; i.e., WT:*lux* + Δ*gndRX* and WT + Δ*gndRX*:*lux*). All cultures were treated with ciprofloxacin for 72 h, and net survival is shown. Data information: In (B, C, E), the bar represents the mean. In (B, C) (right plot), statistical hypothesis testing was performed with the Welch's *t* test (n.s. = not significant; *P* < 0.05). (C) Left plot: statistical hypothesis testing was performed with the Mann–Whitney *U* test (**P* < 0.05). The limit of detection on all applicable plots is indicated with a dashed gray line. Source data are available online for this figure.

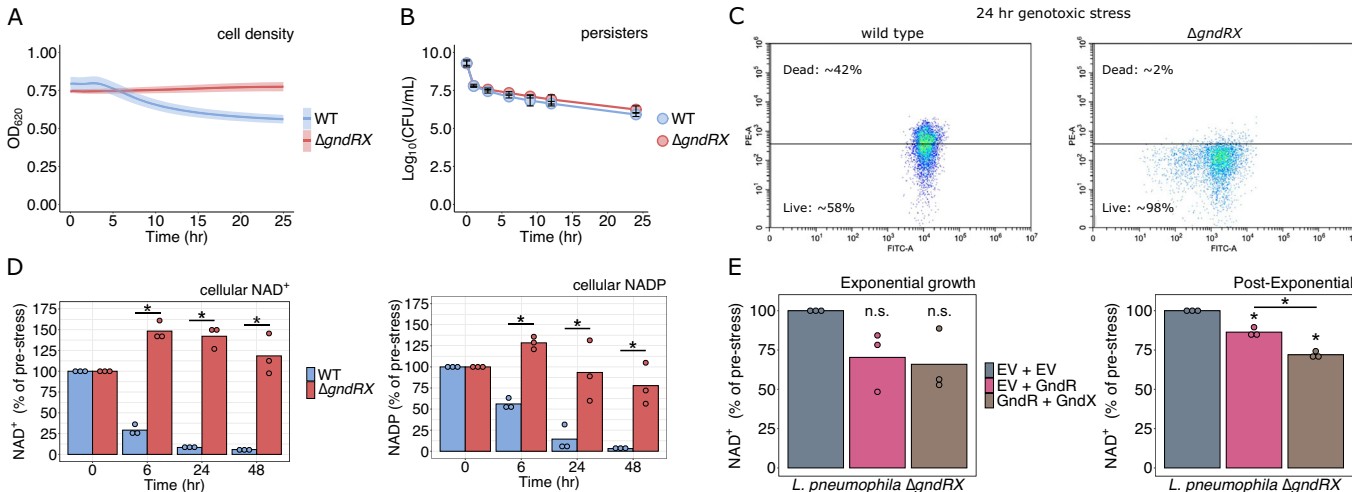

**Figure 7. GndRX appears to shift the cell from a state of dormancy to death during stress.**

(A) Culture turbidity measurements for the wild-type and ΔgndRX strains during the first 24 h of treatment with ciprofloxacin (n = 3 biological replicates). Immediately after the addition of ciprofloxacin, cells were transferred to a flat-bottom 96-well plate (200 µL volume) and absorbance (600 nm) was monitored every 15 min for 24 h. (B) Persister cell counts for the wild-type and ΔgndRX strains during the first 24 h of treatment with ciprofloxacin (n = 2 biological replicates). (C) Flow cytometry quantification using Live/Dead staining for wild-type and ΔgndRX populations after 24 h of ciprofloxacin treatment (10,000 events are displayed; data are representative of n = 3 biological replicates). (D) Quantification of cellular NAD$^+$ (left) and NADP (right) pools over 48 h of ciprofloxacin treatment for wild-type and ΔgndRX strains (n = 3 biological replicates). (E) Quantification of cellular NAD$^+$ pools in ΔgndRX cultures grown to exponential (left) or post-exponential (right) phase. Comparisons are between strains expressing either GndR, GndR + GndX, or an empty vector control (n = 3 biological replicates). Data information: In (A, B), data are presented as the mean (averaged for clarity) ± SEM. (D, E) The bar represents the mean, and statistical hypothesis testing was performed with the Welch's t test (n.s. = not significant; *P < 0.05). (E) Comparisons were made between either GndR or GndR + GndX and the EV control, unless indicated with a horizontal bar. Source data are available online for this figure.

ΔgndRX cultures as early as 6 h post-stress induction, despite the turbidity measurements still increasing during that time. This raises the possibility that both replication and cell death/lysis are occurring in the ΔgndRX population simultaneously.

Given the differential lysis we observed between populations after the induction of genotoxic stress, we wondered what the state of the non-lysed cells in each population was during this time. To investigate this, we quantified cell viability for both strains using flow cytometry and the BacLight LIVE/DEAD staining kit. After 24 h of genotoxic stress, nearly half of the wild-type population was no longer viable and had lost proton motive force across the cell membrane, indicative of dying and damaged cells (Fig. 7C and Appendix Fig. S2C). In contrast to this, cells in the ΔgndRX population almost all retained viability, though we did observe a high variability in DNA-staining intensity by SYTO 9 that is as-yet unexplained (Fig. 7C; x axis of the flow cytometry plots, FITC channel). As there is no difference between the persister counts for these strains at this timepoint and the majority of ΔgndRX cells are no longer culturable (Fig. 7B), this retained viability is consistent with the viable but nonculturable (VBNC) state of dormancy. In this state, L. pneumophila VBNC cells have been shown to retain membrane potential and metabolic activity but are unable to be resuscitated using conventional culturing (Schmid and Hilbi, 2025).

Interestingly, these differences in cell lysis and viability were not accompanied by any consistent or singular changes in morphology (Appendix Fig. S2D), such as filamentation, though the ΔgndRX cells appeared to possess a more heterogeneous distribution of cell shapes and sizes. Given that L. pneumophila VBNC cells are characterized by distinct transformations in their ultrastructure (Al-Bana et al, 2014), an important goal of future work will be to determine whether ΔgndRX

cells adopt similar morphological changes. Finally, we sought to uncover what mechanism was responsible for cell death in these strains. As the RES toxin homologs of GndR all consume or deplete NAD$^+$ to poison the cell, we hypothesized that a similar phenomenon could be occurring in L. pneumophila cells. In support of this, cellular NAD$^+$ and NADP (Fig. 7D) were highly reduced in wild-type cells during genotoxic stress, whereas NAD levels in the ΔgndRX population initially increased following genotoxic stress exposure before eventually declining, though at a much slower rate than that of the wild-type strain. To test whether this was the direct consequence of GndRX, we measured changes in NAD$^+$ with ectopic expression of either GndR or GndR + GndX during unstressed growth. From this, we observed a trend toward reduced NAD$^+$ levels in cells grown to exponential phase and a significant reduction of NAD$^+$ levels in cells grown to post-exponential phase with both GndR and GndRX expression (Fig. 7E). Furthermore, NAD$^+$ levels were more reduced during post-exponential growth when both proteins were expressed. These results suggest that GndRX has the capacity to deplete cellular NAD but the effect of this activity is not growth inhibitory under normal conditions. In summary, the presence or absence of the GndRX system produces distinct and pronounced effects on cell survival and viability under genotoxic stress conditions, with a large proportion of the wild-type population rapidly dying off while the ΔgndRX population transitions to a possible VBNC state.

## Discussion

TA systems are widely abundant in prokaryotic genomes, yet their involvement in bacterial physiology is still poorly understood.

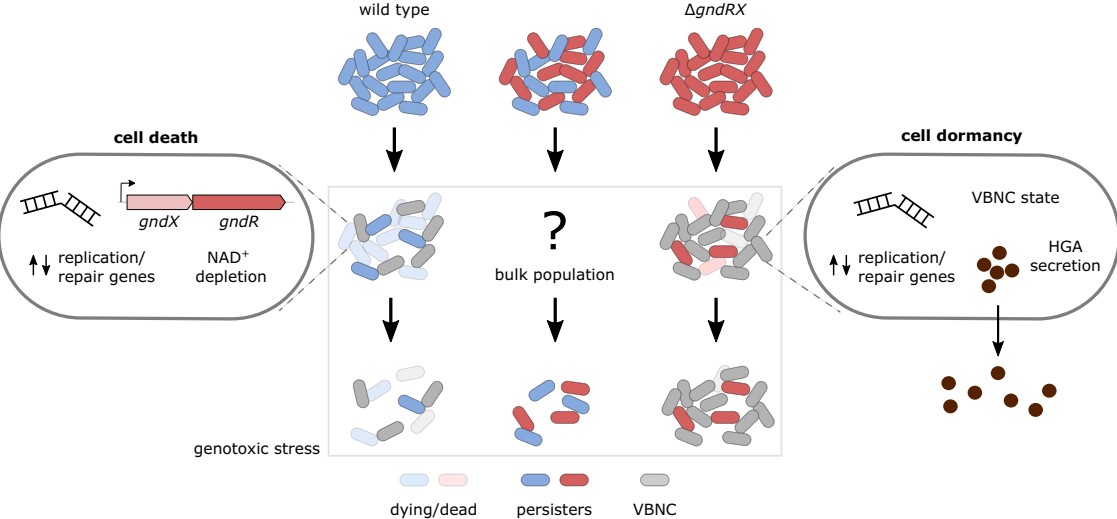

**Figure 8.　Model of GndRX system function during genotoxic stress.**

Overview of GndRX system function during genotoxic stress. After initial stress exposure, *gndRX* transcripts become enriched in wild-type cells, which subsequently undergo rapid NAD$^+$/NADP depletion. The majority of the cells in the population are lysed or become damaged and begin to die, while a subset retain viability but are no longer culturable (VBNC state). Conversely, most Δ*gndRX* cells transition to the VBNC state, which is accompanied by a slower decline in NAD$^+$/NADP levels and secretion of the metabolite HGA. During this time, the wild-type and Δ*gndRX* transcriptomes diverge considerably, including the differential enrichment of transcripts from DNA replication and repair genes. After prolonged stress exposure, wild-type persister cells show highly reduced survival or resuscitation capacity relative to the Δ*gndRX* strain. When the two strains are mixed prior to genotoxic stress exposure, the persister cell counts remain identical throughout the duration of stress exposure.

Despite growing evidence for their role in bacterial immunity, the presence of numerous chromosomal systems in most bacteria suggests an underexplored functional landscape. TA systems have long been hypothesized to act in cellular stress pathways, such as in persistence (Balaban et al, 2004; Rotem et al, 2010) and programmed cell death (Engelberg-Kulka et al, 2005; Nariya and Inouye, 2008). Yet, despite extensive research into the biology of these systems, their involvement in the response to stress remains controversial (Ramisetty et al, 2016b; Song and Wood, 2018; Ronneau and Helaine, 2019; Fraikin et al, 2020; Jurėnas et al, 2022). One intriguing possibility is that bacteria can co-opt TA systems to leverage their molecular activity for non-canonical purposes. As modular and autoregulating elements, there is considerable potential for TA system integration into the cell's genetic circuitry, particularly as the enzymatic activity of a toxin is highly tunable. Such exaptation could therefore involve either tight regulation or complete abrogation of toxin activity, followed by conservation and a resultant atypical signature of vertical inheritance.

Consistent with such a phenomenon, we herein report the discovery and characterization of a conserved TA-like system GndRX in *L. pneumophila* that appears to act specifically in the response to genotoxic stress (Fig. 8). This system lacks detectable toxicity within the cell yet shares remote homology only with RES-Xre TA systems. GndRX is highly conserved both within the genus *Legionella* and across *L. pneumophila* strains, where it is part of the core genome of this species. The deletion of this system has no effect on cell growth, host infection, or survival during translation or cell wall stress, however, under conditions of DNA damage its absence results in a dramatic increase in cell survival. This is quite surprising, given that it is the opposite of what would be expected if TA systems were contributing to persister formation—and because TA system deletions often do not produce phenotypes (Van

Melderen, 2010). In wild-type *L. pneumophila* carrying the GndRX system, DNA stress causes rapid cell death, and over the course of prolonged stress, persister cells die off more rapidly than the Δ*gndRX* strain (Fig. 8). Without this system, most cells do not die following stress exposure and instead transition to the VBNC state. Within *Legionella* biology, VBNC cells are a common developmental form adopted in response to diverse environmental stresses and are critical to the bacterium's capacity for prolonged survival during harsh extracellular conditions (Robertson et al, 2014). Thus, by regulating the transition to either death or the VBNC state, GndRX appears to play an important role in the bacterium's life cycle as opposed to mere selfish parasitism.

Importantly, the positive and rapid enrichment of *gndRX* transcripts following DNA damage induction provides strong evidence for the activation of this system by a specific cue and a high level of integration into the cell's stress response circuitry. The transcriptomic divergence between wild-type and Δ*gndRX* strains during stress also included several genes involved in DNA replication and repair. Among these were the resolvases *ruvA* and *ruvC*, which process the Holliday junctions formed during DNA recombination repair, and the recombination repair gene *recF*. Interestingly, the deletion of *ruvA* has previously been shown to reduce persister survival to fluoroquinolone antibiotics, whereas deletion of *recF* has been reported to increase survival due to the poisoning effects of overactive recombination repair (Theodore et al, 2013; Lemma et al, 2022). These differences might reflect the respective cell states of each strain, with wild-type cells engaging in active yet toxic damage repair while the Δ*gndRX* repair response is stalled. Consistent with this, GO pathways and processes related to cell division, glycolysis, and proton transport were negatively enriched in the Δ*gndRX* strain, suggesting a possible cessation of replication and metabolism, whereas the positive enrichment of

flagellum-related genes in the wild-type may reflect the activation of stress signaling pathways associated with exit from the host into the environment (Robertson et al, 2014; Striednig et al, 2021). Genotoxic stress also triggered the abundant secretion of HGA by the ΔgndRX strain, whereas the production of this metabolite was absent in the wild-type. HGA remains a poorly characterized component of *Legionella* physiology, however roles in nutrient scavenging (Chatfield and Cianciotto, 2007; Zheng et al, 2013) and competitive exclusion (Levin et al, 2019; Holland et al, 2023) have been reported. Importantly, HGA is secreted primarily during late exponential and stationary phase, suggesting it might be associated with stress pathways that are activated prior to exit from the host or during the conversion to a dormant state following nutrient exhaustion. While HGA is not directly implicated in survival, it remains unclear whether its secretion by the ΔgndRX strain is a direct response to DNA damage or instead an indirect consequence of the cell transitioning to the VBNC state.

In addition to how each strain survives DNA stress alone, we observed a striking phenomenon whereby enhanced persister survival can be conferred to wild-type cells from the ΔgndRX strain in a contact-dependent manner. While this may arise from a high local concentration of some signaling molecule, this seems unlikely given the free diffusion of HGA that we observed across the transwell membranes. Furthermore, large membrane pores (0.4 μm) were sufficient to abrogate enhanced survival but are unlikely to prohibit the transit of extracellular material, such as outer membrane vesicles. In support of this, a recent study reported that vesicle sizes from a closely related *L. pneumophila* strain largely fall within a range of 20–120 nm with a median diameter of 62 nm (Fan et al, 2023), and are thus much smaller than the pore sizes we tested. While these findings strongly support a requirement of cell–cell contact, it remains unclear what pathway or process underlies this phenomenon. Contact-dependent communication in bacteria can occur through multiple distinct mechanisms (Blango and Mulvey, 2009; Troselj et al, 2018), including surface protein interactions (such as pilins) and outer membrane exchange. Bacteria can transfer molecular cargo through contact-dependent inhibition and other secretion systems, and cell surface receptors can mediate intercellular recognition and biofilm aggregation. In addition, the exchange of metabolites and cellular material has been reported for intercellular channels (Mullineaux et al, 2008) and membrane-derived nanotubes (Pande et al, 2015), while cell–cell aggregation and biofilm formation can be facilitated by electro-active nanowires in some species (Reguera, 2018). Consequently, there are numerous mechanisms by which bacterial cells can physically interact and communicate. The goal of future work will therefore be to determine the mechanism used by *L. pneumophila* cells to confer enhanced persister survival.

It is critical to note that our findings appear to involve two distinct yet likely interconnected phenomena: the initial progression to cell death or the VBNC state in each bulk population upon stress exposure and the subsequent differential survival/viability of the persister subpopulations after prolonged stress. A link between these two outcomes—and consistent with the contact-dependent survival phenotype—could be that the state of the bulk population over time during DNA stress influences downstream persister survival or resuscitation. For example, in the ΔgndRX population during DNA stress there is reduced cell death resulting in a higher density of VBNC cells. During our co-culture experiments, this

abundance of VBNC cells might influence co-cultured wild-type persisters, through frequent physical interactions, to survive better than they would in an otherwise wild-type population. In this scenario, the health of the broader population would thus inform persister cells as to whether conditions were favorable for resuscitation or alternatively to remain dormant, which manifests as quantifiable colonies on a plate. Consistent with this, reducing the proportion of ΔgndRX cells in the population would reduce the total number of VBNC cells and thereby produce wild-type levels of persister survival, which is precisely what we observed when altering the ratio of wild-type to ΔgndRX cells. Importantly, these density-dependent survival kinetics would not be unprecedented in *L. pneumophila* biology, as cell density has a known effect on resistance to the growth inhibitory effects of HGA (Levin et al, 2019; Holland et al, 2023). Taken together, our work suggests the occurrence of a contact-dependent interrogation of the surrounding bacterial population during stress, which is subsequently integrated into the sensing population's behavior.

While our findings revealed a physiological role for GndRX in the cell, several outstanding questions remain regarding how the system functions. First, it is unclear whether the primary activity of this system causes the depletion of cellular $NAD^+$ and if so, how this is achieved. Given the importance of the R-E-S residues in GndR and its evolutionary relatedness to other RES toxins, it is likely that this protein performs an enzymatic activity which directly depletes or consumes $NAD^+$. The NatT toxin in particular shows considerable similarity to the predicted GndR structure, including the expanded N-terminal flap motif, and thus it is possible that GndR may also function as an NAD phosphorylase. The GndRX system shares many other properties with NatRT, including low or reduced toxicity in its native host cell, the requirement of both proteins for activity, and conservation with the core genome. As with NatRT, the lack of toxicity for GndRX—despite rapidly reducing cellular NAD levels—may be the consequence of a tolerable reduction of this metabolite that is compensated for by NAD salvage pathways. Conversely, NatT toxicity was still observed when this protein was co-expressed in excess relative to NatR and in *E. coli* cells, which we did not observe for GndRX. Furthermore, no deletion phenotype was observed for *natRT* and it functions in the formation of persisters during stress rather than leading to cell death. Thus, while a growing number of RES-Xre TA systems have been characterized in recent years, the functionality of GndRX is sufficiently divergent to argue against its classification as a canonical TA system. Indeed, this system appears to be uniquely non-toxic, is activated by a specific cellular stress, and reduces rather than improves cell survival during stress conditions.

These observations therefore raise the question: why have this system in the first place? The rapidly expanding catalog of bacterial immune systems has revealed both the prevalence of $NAD^+$ depletion as a mode of defense (Boyle and Hatoum-Aslan, 2023) and similar instances of TA-based systems that rely on both proteins for toxic activity (Burman et al, 2024). Perhaps, GndRX may function as one component of a larger pathway and serves an intermediary role in converting a DNA damage signal into the depletion of a critical cellular metabolite, thereby acting as a homeostatic sensor rather than an autonomous immune element. Furthermore, it is also possible that cell death is in fact an exacerbated consequence of the magnitude and duration of genotoxic stress exposure we utilized, and this system would otherwise produce a bacteriostatic effect under physiological

conditions. For example, when experiencing acute DNA stress it may be advantageous for cells to restrict growth to reduce oxidative damage and await improving conditions for repair to proceed. Alternatively, as not all wild-type cells die even after prolonged damage, activation of GndRX may instead be a means of heterogeneity generation and cellular altruism. As vacuolar compartments and biofilm communities are inherently spatially constrained environments with finite resources, it could be advantageous for damaged or genotypically compromised cells to undergo growth restriction or programmed cell death to liberate replicative resources for kin cells. Finally, it is possible that GndRX is indirectly activated upon DNA stress induction and performs a secondary function, which under conditions of prolonged and irreparable genotoxic stress, leads to cell death.

In summary, we sought to investigate the relationship between TA systems and stress tolerance by constructing a pan-TA deletion strain in *L. pneumophila*. Using this strain, we discovered and characterized the GndRX system, which is involved in cell death following genotoxic stress and when deleted, revealed a contact-dependent survival response following prolonged stress. Notably, no other predicted TA systems in *L. pneumophila* influenced cell survival, however the stresses we exposed cells to were not exhaustive and it is possible that others—such as infection within *L. pneumophila*'s natural amoebal hosts or prolonged exposure to nutrient-deprived freshwater environments—would reveal a phenotype for the deletion of another system. Despite its remote homology to several RES-Xre systems and the hypothesized involvement of TA modules in persistence, GndRX reduces persister survival relative to a strain devoid of it. This observation, combined with its apparent lack of toxicity in the cell and high conservation in the *Legionella* genus, suggests that it is a domesticated TA module and may in fact have been co-opted from an ancestral system with canonical activity. Overall, this work provides a unique example of the possible exaptation of a TA system by a bacterium and reveals previously unreported examples of a cell death response and contact-dependent survival mechanism during genotoxic stress. These findings highlight the potential for TA systems to be functionally integrated into the genetic circuitry of the cell, thereby broadening the scope of what these enigmatic elements do in bacteria.

## Methods

### Reagents and tools table

| Reagent/resource | Reference or source | Identifier or catalog number |
|---|---|---|
| **Experimental models** | | |
| *E. coli* TOP10 | Lab stock | JL-E4 |
| *E. coli* TOP10 pBAD33 | This study | JL-E93 |
| *E. coli* TOP10 pBAD33-lpg1604 | This study | JL-E248 |
| *E. coli* TOP10 pBAD33-lpg1605 | This study | JL-E249 |
| *L. pneumophila* Lp01JK | Rao et al, 2013 | JL-L1 |
| *L. pneumophila* Lp01JK + pJB1806 | This study | JL-L26 |
| *L. pneumophila* Lp01JK Δlpg2368-2370 | This study | JL-L29 |
| *L. pneumophila* Lp01JK + pNT562 | This study | JL-L69 |
| *L. pneumophila* Lp01JK Δlpg0488-0489 | This study | JL-L80 |
| *L. pneumophila* Lp01JK Δlpg1604-1605 | This study | JL-L81 |

| Reagent/resource | Reference or source | Identifier or catalog number |
|---|---|---|
| *L. pneumophila* Lp01JK Δlpg1934-1935 | This study | JL-L82 |
| *L. pneumophila* Lp01JK Δlpg2377-2380 | This study | JL-L83 |
| *L. pneumophila* Lp01JK Δlpg2914-2915 | This study | JL-L84 |
| *L. pneumophila* Lp01JK Δlpg2920-2921 | This study | JL-L85 |
| *L. pneumophila* Lp01JK Δ7TA-1 | This study | JL-L86 |
| *L. pneumophila* Lp01JK Δ7TA-2 | This study | JL-L88 |
| *L. pneumophila* Lp01JK::lux | Ensminger et al, 2012 | JL-L99 |
| *L. pneumophila* Lp01JK::lux Δlpg1604-1605 | This study | JL-L100 |
| *L. pneumophila* Lp01JK Δlpg1604-1605 + lpg1604 (chromosomal complement, includes 105 bp upstream of lpg1605) | This study | JL-L104 |
| *L. pneumophila* Lp01JK Δlpg1604-1605 + lpg1605 (chromosomal complement) | This study | JL-L105 |
| *L. pneumophila* Lp01JK Δlpg1604-1605 + lpg1604-05 (chromosomal complement) | This study | JL-L107 |
| *L. pneumophila* Lp01JK Δ7TA-1 (Δlpg1604-05 deletion fixed to add in 105 bp upstream of lpg1605) | This study | JL-L108 |
| *L. pneumophila* Lp01JK Δ7TA-1 + lpg1604-05 (chromosomal complement) | This study | JL-L109 |
| *L. pneumophila* Lp01JK ΔphhA | This study | JL-L187 |
| *L. pneumophila* Lp01JK Δlpg1604-1605 ΔphhA | This study | JL-L189 |
| *L. pneumophila* Lp01JK lpg1604(R84A) | This study | JL-L190 |
| *L. pneumophila* Lp01JK lpg1604(E105A) | This study | JL-L191 |
| *L. pneumophila* Lp01JK lpg1604(S178A) | This study | JL-L192 |
| *L. pneumophila* Lp01JK Δlpg1604-1605 + pNT562 | This study | JL-L201 |
| *L. pneumophila* Lp01JK Δlpg1604-1605 + pNT562-lpg1604-05 | This study | JL-L204 |
| *L. pneumophila* Lp01JK Δlpg1604-1605 + pJB1806 | This study | JL-L240 |
| *L. pneumophila* Lp01JK + pJB1806-lacZ | This study | JL-L241 |
| *L. pneumophila* Lp01JK Δlpg1604-1605 + pJB1806-lacZ | This study | JL-L242 |
| *L. pneumophila* Lp01JK Δlpg1604-1605 + pNT562 + tet-pJB1806 | This study | BN-L868 |
| *L. pneumophila* Lp01JK Δlpg1604-1605 + pNT562 + tet-pJB1806-lpg1605 | This study | BN-L869 |
| *L. pneumophila* Lp01JK Δlpg1604-1605 + pNT562-lpg1604 + tet-pJB1806 | This study | BN-L871 |
| *L. pneumophila* Lp01JK Δlpg1604-1605 + pNT562-lpg1604 + tet-pJB1806-lpg1605 | This study | BN-L872 |
| *L. pneumophila* Lp01JK Δlpg1604-1605 + lpg1605 (chromosomal complement) + pNT562 | This study | BN-L886 |
| *L. pneumophila* Lp01JK Δlpg1604-1605 + lpg1605 (chromosomal complement) + pNT562-lpg1604 | This study | BN-L887 |
| *L. pneumophila* Lp01JK Δlpg1604-1605 + lpg1605 (chromosomal complement) + pNT562-lpg1604(E32D) | This study | BN-L888 |
| *L. pneumophila* Lp01JK lpg1604(E32D) | This study | BN-L898 |
| *Saccharomyces cerevisiae* Y8800 | Yu et al, 2008 | JL-Y1 |
| *Saccharomyces cerevisiae* Y8800 pDEST-AD pDEST-DB | Dreze et al, 2010 | JL-Y32 |
| *Saccharomyces cerevisiae* Y8800 pDEST-AD pDEST-DB-lpg1604 | This study | JL-Y76 |
| *Saccharomyces cerevisiae* Y8800 pDEST-AD-lpg1605 pDEST-DB | This study | JL-Y77 |
| *Saccharomyces cerevisiae* Y8800 pDEST-AD-lpg1605 pDEST-DB-lpg1604 | This study | JL-Y78 |

| Reagent/resource | Reference or source | Identifier or catalog number |
|---|---|---|
| *Saccharomyces cerevisiae* Y8800 pDEST-AD-lpg1604 pDEST-DB | This study | JL-Y79 |
| *Saccharomyces cerevisiae* Y8800 pDEST-AD pDEST-DB-lpg1605 | This study | JL-Y80 |
| *Saccharomyces cerevisiae* Y8800 pDEST-AD-lpg1604 pDEST-DB-lpg1605 | This study | JL-Y81 |
| **Recombinant DNA** | | |
| Empty vector, PBAD promoter, ColE1/pMB1/pBR322/pUC ori | Guzman et al, 1995 | pBAD33 |
| Arabinose-inducible lpg1604 | This study | pBAD33-lpg1604 |
| Arabinose-inducible lpg1605 | This study | pBAD33-lpg1605 |
| For Gateway cloning into pDEST vectors, ColE1/pMB1/pBR322/pUC ori | This study | pDONR221-lpg1604 |
| For Gateway cloning into pDEST vectors, ColE1/pMB1/pBR322/pUC ori | This study | pDONR221-lpg1605 |
| Constitutive expression of GAL4 AD domain, pUC ori | Dreze et al, 2010 | pDEST-AD-ccdb |
| Constitutive expression of GAL4 DB domain, pUC ori | Dreze et al, 2010 | pDEST-DB-ccdb |
| Constitutive expression of GAL4 AD domain fusion with Lpg1604 | This study | pDEST-AD-lpg1604 |
| Constitutive expression of GAL4 AD domain fusion with Lpg1605 | This study | pDEST-AD-lpg1605 |
| Constitutive expression of GAL4 DB domain fusion with Lpg1604 | This study | pDEST-DB-lpg1604 |
| Constitutive expression of GAL4 DB domain fusion with Lpg1605 | This study | pDEST-DB-lpg1605 |
| Empty vector, Ptac promoter, RSF1010 ori | Bardill et al, 2005 | pJB1806 |
| IPTG-inducible lacZ | This study | pJB1806-lacZ |
| pJB1806 with Ptac replaced with tet-inducible promoter, and lac repressor replaced with tet repressor | This study | tet-pJB1806 |
| Tetracycline-inducible lpg1605 | This study | tet-pJB1806-lpg1605 |
| IPTG-inducible lpg1605 | This study | pJB1806-lpg1605 |
| Empty vector, Ptac promoter, ColE1/pMB1/pBR322/pUC ori | Nishida et al, 2017 | pNT562 |
| IPTG-inducible lpg1604 | This study | pNT562-lpg1604 |
| IPTG-inducible lpg1604(E32D) | This study | pNT562-lpg1604(E32D) |
| IPTG-inducible lpg1604-05 | This study | pNT562-lpg1604-05 |
| **Antibodies** | | |
| **Oligonucleotides and other sequence-based reagents** | | |
| GATCATGTCGACTTAAATATTGGTAATTTTAC | This study | JL-P39 (lpg1604_R) |
| TGCAGGGCAAGATAGGTGAAGTTAG | This study | JL-P97 (lpg2368-70_P1_deletion_F) |
| GGCCCAATTCGCCCTATAGTGAGTCGGGGCATGGAACACCAAAAATTAATTA | This study | JL-P98 (lpg2368-70_P2_deletion_R) |
| GGGTTTGCTCGGGTCGGTGGCATATGGATTAACTAATATCCGCAGTCTCAATC | This study | JL-P99 (lpg2368-70_P3_deletion_F) |
| CGTTATCCCGTAGAGATAATCAGCC | This study | JL-P100 (lpg2368-70_P4_deletion_R) |
| ATCCATAATGGATGTATTTTGAGGC | This study | JL-P101 (lpg2368-70_seq_F) |
| CGACTCACTATAGGGCGAATTGGGCCGCTTTCCAGTCGGGAAACCTG | This study | JL-P104 (MazFCass-F) |
| CATATGCCACCGACCCGAGCAAACCCGAAGAAGTTGTCCATATTGGCCAC | This study | JL-P105 (MazFCass-R) |

| Reagent/resource | Reference or source | Identifier or catalog number |
|---|---|---|
| TTAGTTAATCGGGCATGGAACACCAAAAATTAATTA | This study | JL-P106 (lpg2368-70_P5_deletion_R) |
| TTCCATGCCCGATTAACTAATATCCGCAGTCTCAATC | This study | JL-P107 (lpg2368-70_P6_deletion_F) |
| AAGGTTAAGCGTTACAAAACTAATT | This study | JL-P154 (lpg2377-80_P1_deletion_F) |
| GGGTTTGCTCGGGTCGGTGGCATATGTTTTGATTAACTGATATGATGTCTC | This study | JL-P155 (lpg2377-80_P3_deletion_F) |
| TAAGTACAACCAAAATTATAAACCC | This study | JL-P156 (lpg2377-80_P4_deletion_R) |
| TTAATCAAAAGGGCATGGAACACCAAAAATTAATT | This study | JL-P157 (lpg2377-80_P5_deletion_R) |
| TTCCATGCCCTTTTGATTAACTGATATGATGTCTC | This study | JL-P158 (lpg2377-80_P6_deletion_F) |
| ATCACTTTCAACTTGTTCTTT | This study | JL-P159 (lpg2377-80_seq_F) |
| AACCCACTCTACGAACTAGCT | This study | JL-P160 (lpg2377-80_screen_F) |
| AATAGTAAAACAAATGGCCAT | This study | JL-P161 (lpg2377-80_screen_R) |
| TCTAATATAAATGCAGTTGGGTTTC | This study | JL-P168 (lpg0488-89_P1_deletion_F) |
| GGCCCAATTCGCCCTATAGTGAGTCGTTTCATTGAGCATAAGTCTAAGCAT | This study | JL-P169 (lpg0488-89_P2_deletion_R) |
| GGGTTTGCTCGGGTCGGTGGCATATGAGCTGATTAGATCTCATGGATGCCG | This study | JL-P170 (lpg0488-89_P3_deletion_F) |
| CGTTTTGTGAATATACAGGATCAGG | This study | JL-P171 (lpg0488-89_P4_deletion_R) |
| CTAATCAGCTTTTCATTGAGCATAAGTCTAAGCAT | This study | JL-P172 (lpg0488-89_P5_deletion_R) |
| CTCAATGAAAAGCTGATTAGATCTCATGGATGCCG | This study | JL-P173 (lpg0488-89_P6_deletion_F) |
| GCATCCAGGAGATGGTCAACTT | This study | JL-P174 (lpg0488-89_seq_F) |
| CTAAAAAAAGCGAATACGAGCTGAT | This study | JL-P175 (lpg1604-05_P1_deletion_F) |
| TTAGTTTTACAATAACATCGCCACT | This study | JL-P176 (lpg1604-05_P4_deletion_R) |
| ATCAATGTATATTTAAGATATTTTTCCCAAATTGTG | This study | JL-P177 (lpg1604-05_P5_deletion_R) |
| TATCTTAAATATACATTGATCTGACATGCAGAATA | This study | JL-P178 (lpg1604-05_P6_deletion_F) |
| TGCAAAGACGAGTAGATAAGACAGT | This study | JL-P179 (lpg1604-05_seq_F) |
| CCTGCCTTAGTGAGGTTTTTAAATT | This study | JL-P180 (lpg1934-35_P1_deletion_F) |
| TGCTATCCACAGATAATTTGACAGG | This study | JL-P181 (lpg1934-35_P4_deletion_R) |
| TTATATGGCTCTCTAGCGCCTTTCCTCATTATTTC | This study | JL-P182 (lpg1934-35_P5_deletion_R) |
| GGCGCTAGAGAGCCATATAATCCTACTAAAGTAGA | This study | JL-P183 (lpg1934-35_P6_deletion_F) |
| CCAAGCTAATGTCAGTTAATGTTAT | This study | JL-P184 (lpg1934-35_seq_F) |
| AGGGATAAATATTTGCCATAAGTTA | This study | JL-P185 (lpg2914-15_P1_deletion_F) |
| AAGCTTTATAAATCATAGGTAACGT | This study | JL-P186 (lpg2914-15_P4_deletion_R) |

| Reagent/resource | Reference or source | Identifier or catalog number |
|---|---|---|
| AGCAATGATTGGTTGAT AATTGATGTTTTAACTTT | This study | JL-P187 (lpg2914-15_P5_deletion_R) |
| ATTATCAACCAATCATT GCTATTCTTATTTTTCTT | This study | JL-P188 (lpg2914-15_P6_deletion_F) |
| ATTTATTATCTAGCC ATCAGAATAC | This study | JL-P189 (lpg2914-15_seq_F) |
| GTTACAAAAAACAA TCCGCTCAAAA | This study | JL-P190 (lpg2920-21_P1_deletion_F) |
| CATTGGCTTATTAGA TAAACCGGAT | This study | JL-P191 (lpg2920-21_P4_deletion_R) |
| AGTCCTACGAGGTCATA ACTTACTTACTCCAGATT | This study | JL-P192 (lpg2920-21_P5_deletion_F) |
| AGTTATGACCTCGTAG GACTTTAGGGTCTGTTGAC | This study | JL-P193 (lpg2920-21_P6_deletion_F) |
| GTTATGGCCTTAAAGTA CTTGTAAA | This study | JL-P194 (lpg2920-21_seq_F) |
| GGCCCAATTCGCCCTAT AGTGAGT CGATTTAAGATATTTTTCC CAAATTGTG | This study | JL-P198 (lpg1604-05_P2_deletion_R) |
| GGGTTTGCTCGGGTC GGTGGCATATGATACATTG ATCTGACATGCAGAATA | This study | JL-P199 (lpg1604-05_P3_deletion_F) |
| GGCCCAATTC GCCCTATAGTGAGTCGCTCTAG CGCCTTTCCTCATTATTTC | This study | JL-P200 (lpg1934-35_P2_deletion_R) |
| GGGTTTGCTCGG GTCGGTGGCATATGAGCC ATATAATCCTACTAAAGTAGA | This study | JL-P201 (lpg1934-35_P3_deletion_F) |
| GGCCCAATTCGCCCTAT AGTGAGTCGGGTTGATAATTGA TGTTTTAACTTT | This study | JL-P202 (lpg2914-15_P2_deletion_R) |
| GGGTTTGCTCGGGT CGGTGGCATATGAATCATTGC TATTCTTATTTTTCTT | This study | JL-P203 (lpg2914-15_P3_deletion_F) |
| GGCCCAATT CGCCCTATAGTGAGTCGGGT CATAACTTACT TACTCCAGATT | This study | JL-P204 (lpg2920-21_P2_deletion_R) |
| GGGTTTGCTCG GGTCGGTGGCATATGTCG TAGGACTTTAGGG TCTGTTGAC | This study | JL-P205 (lpg2920-21_P3_deletion_F) |
| GCCCTTGCCTATTACAGGTTAC | This study | JL-P215 (lpg1604_screen_F) |
| CCAATTAC TCCACAGCCTGCAC | This study | JL-P216 (lpg1604_screen_R) |
| CTACGCAACTTTTTGAGCCCAG | This study | JL-P217 (lpg1934_screen_F) |
| CGGTTGTATTGGCAGAAAAATAG | This study | JL-P218 (lpg1934_screen_R) |
| CCATACTGCTTGCCAAAAGCAC | This study | JL-P219 (lpg2914_screen_F) |
| CAATGGCCAAATAAACAGGCCAC | This study | JL-P220 (lpg2914_screen_R) |
| GTACAAAAACGCTTTTGGCATATC | This study | JL-P221 (lpg2920_screen_F) |
| GGTGTGCTCGGAGTCATTTTAG | This study | JL-P222 (lpg2920_screen_R) |
| GATCATGGTACCTAT TAGGGAGGT ATAATATGTATTCCTATATA GATTTTAATGATAAAG | This study | JL-P287 (lpg1604_F) |
| AGGTTATTAGGAT ATTTTTCCCAAATTGTG GCTAGATGAC | This study | JL-P298 (lpg1604_P5_complement_R) |
| GAAAAATATCCTAATAACC TCGCCAAAAGTCCAGAAACTT (lpg1604_P6_complement_F) | This study | JL-P299 |
| TAGTATGCAAATT TAAGATATTTTTCCCAAATTGTG | This study | JL-P316 (lpg1604_P5_del_new_R) |

| Reagent/resource | Reference or source | Identifier or catalog number |
|---|---|---|
| TATCTTAAATTTG CATACTAAACTCCTGACTGCTTTC | This study | JL-P317 (lpg1605_P6_del_new_F |
| GGAGTTTAGTATGTAT TCCTATATAGATTTT AATGATAAAGTGC | This study | JL-P318 (lpg1604_P5_comp_new_R) |
| AGGAATACATACT AAACTCCTGACTGCTTT CTACTATAG | This study | JL-P319 (lpg1604_P6_comp_new_F) |
| GGTTCGCTCTGTATTG TATCAATCATATTG | This study | JL-P328 (lpg1604-05_seq_R) |
| GATCATGGTACC TATTAGGGAGGTATAA TATGCAAACTAATACAAGGTC TTTACAAAGC | This study | JL-P331 (lpg1605_F) |
| GATCATGTCGACCT AATAACCTC GCCAAAAGTCCAG | This study | JL-P334 (lpg1605_R) |
| GAGCACCGCA GAAACGCTTGCAACGC | This study | JL-P345 (phhA_P1_deletion_F) |
| GGCCCAATTCGC CCTATAGTGAGTCGTGTTAATTT GTTATTTTGTAAGACAGCTGG | This study | JL-P346 (phhA_P2_deletion_R) |
| GGGTTTGCTCGG GTCGGTGGCATAT GCTCCATAATCTCTCCTTCACT CATCTCATTC | This study | JL-P347 (phhA_P3_deletion_F) |
| GGGGTTTTTTTTTA CAGACTTAGGC | This study | JL-P348 (phhA_P4_deletion_R) |
| GATTATGGAGTGTTAATT TGTTATTTTGTAAGACAGCTGG | This study | JL-P349 (phhA_P5_deletion_R) |
| CAAATTAACACTCCATA ATCTCTCCTTCACTCATCTCATTC | This study | JL-P350 (phhA_P6_deletion_F) |
| GGCTAAAGCCTGCTC ACCACTTTC | This study | JL-P351 (phhA_screen_F) |
| GGGTATCTCAAAAA GATAATGGTAATCATG | This study | JL-P352 (phhA_screen_R) |
| CCAATCAGTCAATGAACA AACTGTGTTGG | This study | JL-P353 (phhA_seq_F) |
| TGAATCAGCTTTTA GTGATGGCAGTTTTGGT ATTTATTATGC | This study | JL-P358 (lpg1604_P5_R84A_R) |
| CATCACTAAAAG CTGATTCAAATCCAGTATGTG TAAAAGCAGC | This study | JL-P359 (lpg1604_P6_R84A_F) |
| AATAAAGGCTAC TTGCTTTCATCGAGAACG GTTTTATAGC | This study | JL-P360 (lpg1604_P5_E105A_R) |
| GAAAGCAAGTAGCCT TTATTGCTGTCTCAAG AGATGAAGCGGC | This study | JL-P361 (lpg1604_P6_E105A_F) |
| ATATCCAGCTGT AAGAGATTTAAATGGCT TATGTGTGGCTG | This study | JL-P362 (lpg1604_P5_S178A_R) |
| AATCTCTTACAG CTGGATATAGCAAGCCCCAT TCCTTCTTTTC | This study | JL-P363 (lpg1604_P6_S178A_F) |
| ggggacaagtttgtacaaa aaagcaggcttcATGTATTCCTA TATAGATTTTAATG | This study | JL-P500 (attB1_lpg1604_for) |
| ggggaccactttgtacaaga aagctgggtcTTAAATATTGGT AATTTTACTTTC | This study | JL-P501 (attB2_lpg1604_rev) |
| ggggacaagtttgtacaaaaa agcaggcttcATGC AAACTAATACAAGGTC | This study | JL-P502 (attB1_lpg1605_for) |
| ggggaccactttg tacaagaaagctgggtcCTAATAAC CTCGCCAAAAG | This study | JL-P503 (attB2_lpg1605_rev) |
| catgGTCGACaggaggtat aatATGGTCGTTTTACA ACGTCGTGAC | This study | RBS-lacZ Fw Sal |

| Reagent/resource | Reference or source | Identifier or catalog number |
|---|---|---|
| ctcGCATGCCTATT TTTGACACCAGACCAACTG | This study | lacZ Rev SphI |
| gagtcctctctaagctctatagg | This study | 1604E32D P1 |
| gacaaattg agttggacaatcc | This study | 1604E32D P4 |
| GAGCTCGTCTGCAGAATCTGC | This study | 1604E32D P5 |
| GCAGACGAGCT CGAACAAATAGC | This study | 1604E32D P6 |
| cctgATCGATATGcaaac taatacaaggtctttacaaagc | This study | Lpg1605 Fw ClaI |
| ccaggatccctaat aacctcgccaaaagtcc | This study | Lpg1605 Rev BamHI |
| **Chemicals, enzymes, and other reagents** | | |
| N-(2-acetamido)-2-aminoethanesulfonic acid | Bio Basic | AD004 |
| Yeast extract | Gibco | 212750 |
| LB broth (Miller) | BioShop | LBL407.1 |
| Breathe-Easy membrane | Diversified Biotech | BEM-1 |
| Breathe-Easier membrane | Diversified Biotech | BERM-2000 |
| RPMI 1640 medium | Gibco | 11875093 |
| Transwell plate inserts (0.1) | VWR | 10769-176 |
| Transwell plate inserts (0.4) | VWR | 76313-902 |
| chlorophenol red-β-D-galactopyranoside | Sigma-Adrich | 10884308001 |
| NAD-Glo assay kit | Promega | G9071 |
| NADP-Glo assay kit | Promega | G9081 |
| BacLight Live/Dead kit | Invitrogen | L34856 |
| Machery-Nagel Tissue kit | Machery-Nagel | 740952.250 |
| Nextera Library Prep kit | Illumina | FC-131-1024 |
| PureLink RNA extraction kit | Invitrogen | 12183018A |
| **Software** | | |
| Rstudio | https://posit.co/products/open-source/rstudio/ | |
| Inkscape | https://inkscape.org/ | |
| **Other** | | |
| UV crosslinker | Stratagene | Stratalinker 1800 |
| Plate reader | TECAN | Infinite 200 PRO Mplex |
| Growth curve robot | S&P Robotics | |
| Flow cytometer | Beckman | CytoFLEX S |
| Sequencing platform | Illumina | MiniSeq |

## Strains and plasmids

Strains, plasmids, and oligonucleotides used in this study are listed in the Reagents and Tools Table. *L. pneumophila* strains used were derived from Lp01[JK] (Rao et al, 2013). *E. coli* TOP10 cells (Invitrogen) were used for cloning, plasmid maintenance, and in vivo toxicity assays. Bacterial expression plasmids were constructed using PCR products amplified from Lp01[JK] genomic DNA with restriction cloning. Yeast two-hybrid plasmids and strains were constructed as described previously (Lin et al, 2023). All generated constructs were confirmed by Sanger sequencing. Plasmids were introduced into *E. coli* via heat-shock

transformation and into *L. pneumophila* via electroporation. The endogenous TA loci in *L. pneumophila* were deleted and edited using the MazF recombineering method (Bailo et al, 2019) with minor modifications as described previously (Nicholson et al, 2022) to produce scar-free, in frame deletions. Briefly, linear DNA containing a *mazF*-Kan[R] cassette flanked by 2–3 kb chromosomal homology arms is introduced via natural transformation into the desired strain and cells with the cassette integrated via homologous recombination are selected by plating on media containing kanamycin. The integrant strains are next transformed with linear DNA containing homology arms with the desired deletion, and homologous recombination leads to excision of the cassette and counterselection via MazF expression. Cells in which the cassette is lost are detected by identifying those with kanamycin sensitivity. Genomic complementation was performed as above, by replacing the MazF cassette integrant with the desired final edit. Luminescent *L. pneumophila* strains bearing the lux cassette were constructed by introducing genomic DNA, isolated from a previously constructed luminescent strain (Ensminger et al, 2012) via natural transformation into wild-type and ΔgndRX strains. Recombination and integration of the lux cassette were selected for by plating on kanamycin-containing media and luminescence was confirmed visually. All strains and plasmids are available upon request.

## Media and culture conditions

Bacterial experiments and routine strain maintenance were performed at 37 °C. Natural transformation of *L. pneumophila* was performed at 30 °C. *L. pneumophila* strains were grown in *N*-(2-acetamido)-2-aminoethanesulfonic acid (ACES)-buffered yeast extract and on charcoal AYE (CYE) agar plates supplemented with 0.4 g/L L-cysteine and 0.25 g/L ferric pyrophosphate. For liquid growth, cultures were inoculated from patches grown for 2 days. When required for selection or plasmid maintenance, media were supplemented with chloramphenicol (5 μg/mL) or kanamycin (40 μg/mL). Ectopic gene expression was induced by isopropylthio-β-galactoside (IPTG; 100 μM) and repressed with glucose (1% v/v). *E. coli* strains were grown on lysogeny broth (LB, Miller) liquid media and agar. When required, media were supplemented with ampicillin (100 μg/mL), chloramphenicol (34 μg/mL) or kanamycin (40 μg/mL) for selection or plasmid maintenance. Yeast experiments and routine strain maintenance were performed at 30 °C. *S. cerevisiae* strains were grown on yeast peptone adenine dextrose (YPAD) medium (2% bacto peptone w/v, 1% yeast extract w/v, 2% glucose v/v, 180 mg/L adenine sulfate), or synthetic defined (SD) medium comprised of yeast nitrogen base with ammonium sulfate, supplemented with 2% glucose and all amino acids, lacking specific amino acids where necessary for selection or plasmid maintenance.

## In vivo bacterial toxicity experiments

Growth assays were performed as follows: freshly struck *E. coli* and *L. pneumophila* strains containing plasmids expressing the genes of interest were grown overnight in the presence of 1% glucose. Cultures were washed to remove glucose and adjusted to $OD_{600} = 0.1$ in fresh media supplemented with either arabinose (0.2%) or the indicated concentrations of anhydrotetracycline (aTc) and/or

IPTG. Cultures for standard growth assays were plated in triplicate in a flat-bottom 96-well plate (100 µL volume for *E. coli*, 200 µL volume for *L. pneumophila*), sealed with a Breathe-Easy sealing membrane (Diversified Biotech BEM-1), and absorbance (600 nm) was monitored every 15 min for 24 h using an S&P growth curve robot. Cultures for co-expression titration growth assays were plated in a flat-bottom 96-well plate at 200 µL per well, sealed with a Breathe-Easier sealing membrane (Diversified Biotech BERM-2000) and incubated for 24 h at 37 °C without shaking. After incubation, the plate seal was removed and the absorbance (600 nm) for each well was measured using a TECAN plate reader.

## Yeast two-hybrid assays

Yeast two-hybrid (Y2H) experiments were performed as described previously (Lin et al, 2023). Briefly, proteins of interest were fused to either the GAL4 transcriptional activating domain (AD) or DNA-binding domain (DB), using the pDEST-AD-ccdB and pDEST-DB-ccdB constitutively active Gateway destination plasmids. Three independent clones of Y8800 containing pDEST-AD and pDEST-DB encoding gene fusions of interest were grown overnight at 30 °C in liquid SD -Leu/Trp media supplemented with 2% glucose. These cultures were then stamped on plates containing histidine (control) or lacking histidine (physical interaction selection). Plates were imaged after 2 days of growth at 30 °C.

## U937 growth assays

Intracellular growth assays were performed as described previously (Lin et al, 2023). Briefly, post-exponential (OD$_{600}$ = 3.6–4) *L. pneumophila* cells were used to infect duplicate monolayers of differentiated U937 monocytes (American Tissue Culture Collection, CRL-1593.2) at a multiplicity of infection of 0.1 and incubated at 37 °C with 5% CO$_2$ (RPMI 1640 medium, Gibco). After 2 h, cells were washed three times with RPMI and resuspended in fresh RPMI and infections were carried out for 72 h. At each indicated timepoint post infection, U937 cells were lysed with 0.02% saponin and bacteria were plated on CYE agar supplemented with streptomycin to quantify colony-forming units (CFUs).

## Antibiotic disk diffusion assays and dose–response curves

For antibiotic disk diffusion assays, overnight cultures of *L. pneumophila* were grown to the exponential phase (OD$_{600}$ = 2) and plated as a lawn on CYE agar plates. A sterile disk was then placed in the middle of the plate and impregnated with antibiotics at the indicated concentrations. Plates were incubated at 37 °C for 96 h and imaged. For antibiotic dose–response curves, overnight cultures of *L. pneumophila* were grown to exponential phase (OD$_{600}$ = 2) and sub-cultured to OD$_{600}$ = 0.1 with twofold serial dilutions of antibiotics at the indicated concentrations. Cultures were incubated in a flat-bottom 96-well plate (200 µL volume) at 37 °C, sealed with a Breathe-Easy sealing membrane (Diversified Biotech), and absorbance (600 nm) was monitored every 15 min for 24 or 48 h using an S&P growth curve robot. Relative fitness was calculated as the percent growth (area under the curve) relative to a control with no antibiotics added. Following incubation for the mitomycin C dose–response assay, cultures were washed twice with sterile PBS and tenfold serial dilutions were plated

onto CYE agar plates supplemented with streptomycin. After 4 days of growth, CFU values were calculated and compared to the 0-h timepoint CFUs, which were determined by plating immediately prior to the addition of mitomycin C.

## Minimum inhibitory concentration (MIC) determination

Overnight cultures of *L. pneumophila* were diluted to a concentration of $5 \times 10^6$ cells/mL in fresh media supplemented with twofold serial dilutions of ciprofloxacin or mitomycin C. Cultures were plated in a flat-bottom 96-well plate at 200 µL per well, sealed with a Breathe-Easier sealing membrane (Diversified Biotech BERM-2000), and incubated for 48 h at 37 °C without shaking. After incubation, the plate seal was removed and absorbance (600 nm) for each well was measured using a TECAN plate reader to determine MIC (the lowest drug concentration with no observable growth).

## Persistence time-kill assays

Overnight cultures of *L. pneumophila*, made from fresh patches and streaks, were grown to the exponential phase (OD$_{600}$ = 2) in 125 mL glass flasks containing either 15 mL or 25 mL culture volume. Individual experiments were performed either with the 15 mL culture or the 25 mL culture was split into three technical replicates (6–7 mL) in a 24-deep well plate and covered with Breathe-Easier membranes. Antibiotics were added at the following concentrations: 15 µg/mL gentamicin, 100 µg/mL carbenicillin, 25 µg/mL ciprofloxacin, and 10 µg/mL mitomycin C. Cultures were incubated at 37 °C for the indicated times. CFUs were plated on CYE agar plates supplemented with streptomycin prior to and every 24 h post antibiotic addition, unless otherwise indicated. Cultures treated with fluoroquinolone antibiotics were washed three times with sterile PBS prior to plating to remove cell-bound drug. For the experiments where it is indicated, the culture supernatant was removed, and HGA production was measured by quantifying absorbance at 400 nm. CFU counts below the limit of detection were substituted with a default count of 1.

## UV irradiation growth curves

Experiments to test survival following UV irradiation were performed as described previously (Charpentier et al, 2011), with minor modifications. Briefly, *L. pneumophila* cultures were made from fresh patches and streaks and grown to the exponential phase overnight. The following day, 2 OD units were pelleted and resuspended in distilled water, then aliquoted (250 µL) onto a sterile petri dish (without lid) and irradiated with the indicated doses in a Stratagene Stratalinker. Following irradiation, cultures were pelleted, resuspended in fresh AYE, and adjusted to OD$_{600}$ = 0.1. Growth curves were performed as described above.

## Co-culture and transwell survival experiments

Overnight cultures of *L. pneumophila* strains (with and without the *lux* cassette)—prepared as described above—were mixed 1:1, and time-kill assays were performed. Plated CFUs were imaged with and without external light to distinguish luminescent from non-luminescent cells. For the transwell experiments, 2.5 mL of each strain was added to the top or bottom compartment of a six-well plate containing a 0.1- or 0.4-µm

membrane plate insert (VWR). In each combination, one strain carried the *lux* cassette to ensure that no mixing occurred across the membrane. Ciprofloxacin was added to each compartment, and plates were wrapped in parafilm to prevent evaporation and incubated with shaking at 150 rpm in a plate shaking incubator. CFUs were quantified prior to and 48 h post antibiotic addition (due to the small culture volumes) from each compartment individually. Mixing controls were constructed by poking holes in the membrane with a sterile pipette tip prior to adding cells.

## Conditioned media transplant experiments

Individual cultures of wild-type and Δ*gndRX L. pneumophila* were grown overnight, as described above. Wild-type and Δ*gndRX* cultures were then mixed 1:1 and treated with ciprofloxacin for 24 h, after which point the co-culture supernatant (conditioned media) was harvested and filter sterilized with a 0.2-μm membrane. The sterilized conditioned media were then transplanted into cultures containing each individual strain, which had also been treated with ciprofloxacin for 24 h, and incubated for an additional 48 h (72 h total drug treatment).

## Cell lysis assays

Cell lysis was measured by quantifying free LacZ in the culture supernatant using a previously described colorimetric reporter assay (Gordils-Valentin et al, 2024), with some modifications. Briefly, wild-type and Δ*gndRX* strains of *L. pneumophila* were transformed with either a plasmid vector (pJB1806) containing the *lacZ* gene under the Ptac promoter or an empty vector control. Overnight cultures of these strains were prepared as described above. Ciprofloxacin and IPTG (500 μM) were added during the exponential phase and incubated for 24 h. At the indicated time points, the supernatant was removed, mixed with the LacZ substrate chlorophenol red-β-D-galactopyranoside (CPRG; 20 μg/mL; Sigma), and incubated for 24 h at room temperature. LacZ hydrolysis of CPRG was quantified by measuring absorbance at 575 nm.

## Cellular NAD quantification

In vivo pools of $NAD^+$ and NADP were quantified using the NAD-Glo and NADP-Glo assays (Promega) per the manufacturer's instructions and with some modifications. Briefly, overnight cultures of wild-type and Δ*gndRX L. pneumophila* strains were grown to the exponential phase as described above and treated with ciprofloxacin. At the indicated time points, cells were washed twice in PBS and resuspended at $OD_{600} = 0.5$ in PBS. Cells were assayed in technical duplicate and incubated in flat-bottom white tissue culture plates. Luminescence was measured using a TECAN plate reader every 5 min for 1 h at room temperature. To determine the effect of overexpressing GndRX on $NAD^+$ levels, the indicated strains were grown in the presence of 250 μM IPTG until they reached exponential ($OD_{600} = \sim 2$) or post-exponential ($OD_{600} = \sim 4$) growth phase, and $NAD^+$ levels were determined as described above.

## Flow cytometry

Cell viability prior to and after antibiotic-induced genotoxic stress was measured using the BacLight Live/Dead kit (Invitrogen) per the manufacturer's instructions and as described previously (Braun

et al, 2019), with some modifications. At the indicated time points, $1 \times 10^7$ *L. pneumophila* cells were pelleted and resuspended in 0.85% NaCl. Cells were stained for 15 min at RT in the dark with the Live/Dead staining solution, then pelleted and resuspended in NaCl. Control cells that were unstained or heat-killed were also quantified. Cells were then run on a Beckman CytoFLEX S flow cytometry platform to determine live/dead proportions.

## Genome sequencing

*L. pneumophila* genomic DNA was extracted using a Machery-Nagel Tissue kit and prepared for next-generation sequencing using a Nextera Library prep kit per the manufacturer's instructions. Genomic libraries from each strain were multiplexed and sequenced on an Illumina MiniSeq platform with a minimum of 50× coverage. Raw sequencing reads were demultiplexed, trimmed using trimmomatic (Bolger et al, 2014), and quality was assessed using FastQC (https://www.bioinformatics.babraham.ac.uk/projects/fastqc/). Reads were de novo assembled using SPAdes (Bankevich et al, 2012) and aligned to the previously sequenced Lp01 "core" genome (the progenitor of Lp01[JK]) (Rao et al, 2013) using Mauve (Darling et al, 2004) to confirm that no genome rearrangements had occurred. Reads were also reference assembled to the Lp01 "core" genome using Bowtie 2 (Langmead and Salzberg, 2012) and single-nucleotide polymorphisms (SNPs) were detected using VCF.Filter (Müller et al, 2017). Gene coverage was determined using Bedtools (Quinlan and Hall, 2010) and read mapping for each gene deletion was confirmed using the Integrated Genome Browser Viewer (Thorvaldsdóttir et al, 2013). To test for sample contamination, unmapped reads were assembled with SPAdes and used as queries in a BLAST (Altschul et al, 1990) search against the NCBI nr database.

## RNA sequencing

RNA was recovered using a PureLink RNA extraction kit (Invitrogen) and sequenced, after rRNA depletion, at the Microbial Genome Sequencing Center (MiGS). Raw sequencing reads were trimmed with trimmomatic and quality checked using FastQC. Filtered reads were aligned to a reference genome (Refseq ID: GCF_001941585.1) using STAR (Dobin et al, 2013) and mapped reads were quantified using featureCounts (Liao et al, 2014). Differential expression analysis was performed with edgeR (Robinson et al, 2010), principal component plots were constructed with DESeq2 (Love et al, 2014), and gene set enrichment analysis was conducted with clusterProfiler (Yu et al, 2012) using gene ontology (GO) annotations retrieved from QuickGO (Binns et al, 2009). Genes were considered to be differentially enriched if they showed a log2 fold change >= [1] and a statistically significant difference in transcript counts between samples ($P < 0.01$, exact test implemented in edgeR, Benjamini and Hochberg (Benjamini and Hochberg, 1995) false discovery rate correction for multiple hypothesis testing).

## TA system conservation across *Legionella* species and *L. pneumophila* strains

A *Legionella* species core genome phylogeny was constructed and orthologous protein groups identified as described previously (Lin et al, 2023). The conservation of all homologs of the seven TA

systems found in the *L. pneumophila* genome across *Legionella* species was plotted on the resulting phylogeny using the R package ggtree (Yu et al, 2017). All complete *L. pneumophila* genomes (117 total) were downloaded from the NCBI Refseq database (O'Leary et al, 2016) in August, 2024. Open-reading frames (ORFs) were annotated using Prodigal (Hyatt et al, 2010) and orthologous groups of ORFs were identified using OrthoFinder (Emms and Kelly, 2019). TA system conservation was then determined for each of the seven systems predicted in the Philadelphia-1 genome.

## GndR homolog search and phylogeny construction

To identify protein homologs of GndR, we queried the UniProtKB database (UniProt Consortium, 2023) using Jackhmmer (5 iterations; default parameters) on the HMMER webserver (Potter et al, 2018). This detected 3995 homologous sequences, which were then clustered using MMseqs2 (80% identify; 80% coverage) (Steinegger and Söding, 2017) into 1090 representative sequences. Taxonomic annotations were retrieved for all sequences using MMseqs2, and a multiple sequence alignment of the representative sequences was constructed using MAFFT (Katoh and Standley, 2013). A protein sequence phylogeny was then constructed using FastTree (Price et al, 2010) and plotted with taxonomic annotations (lowest common ancestor, MMseqs2) using ggtree (Yu et al, 2017). Remote homolog searches for proteins related to GndR were performed using the HHpred (Söding et al, 2005) and Foldseek (van Kempen et al, 2023) webservers with default settings and databases. For Foldseek, we used as a query the predicted structure of GndR (AF-Q5ZV38-F1-v4) from the AlphaFold Protein Structure Database (Varadi et al, 2024). Toxin homolog sequences, along with GndR, were aligned with MUSCLE (Edgar, 2022), and the gene phylogeny was plotted using ggtree.

## Statistical analyses

Data normality was tested using the Shapiro–Wilk test implemented using the R function shapiro.test(). If data satisfied the assumption of normality, statistical hypothesis testing was performed using Welch's *t* test implemented using the R function t.test(). Otherwise, testing was performed using the Mann–Whitney *U* test implemented using the R function wilcox.test().

## Data availability

DNA sequencing data have been deposited in the SRA under the BioProject accession PRJNA1168146. RNA sequencing data have been deposited in the NCBI Gene Expression Omnibus under the accessions GSE278701 (genotoxic stress experiments) and GSE278702 (growth phase experiments).

The source data of this paper are collected in the following database record: biostudies:S-SCDT-10_1038-S44319-025-00545-y.

## Peer review information

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

## Acknowledgements

We thank John MacPherson for performing the Y2H cloning and experiments and José Santé for his help in establishing the LacZ lysis assay. We thank members of the Ensminger lab for helpful discussions and comments on the manuscript. Illumina RNA-seq was performed by the Microbial Genome Sequencing Center (Pittsburgh, USA). Routine plasmid sequencing was performed at The Centre for Applied Genomics, Hospital for Sick Children (Toronto, Canada). JDL was supported by an Ontario Graduate Scholarship. This work was supported by the Natural Sciences and Engineering Research Council of Canada, Grants RGPIN-2020-06636 and RGPAS-2020-00014 (AE).

## Author contributions

**Jordan D Lin**: Conceptualization; Data curation; Formal analysis; Funding acquisition; Investigation; Visualization; Methodology; Writing—original draft; Writing—review and editing. **Beth Nicholson**: Validation; Investigation; Methodology; Writing—review and editing. **Alexander W Ensminger**: Conceptualization; Supervision; Funding acquisition; Visualization; Methodology; Project administration; Writing—review and editing.

Source data underlying figure panels in this paper may have individual authorship assigned. Where available, figure panel/source data authorship is listed in the following database record: biostudies:S-SCDT-10_1038-S44319-025-00545-y.

## Disclosure and competing interests statement

The authors declare no competing interests.

# Expanded View Figures

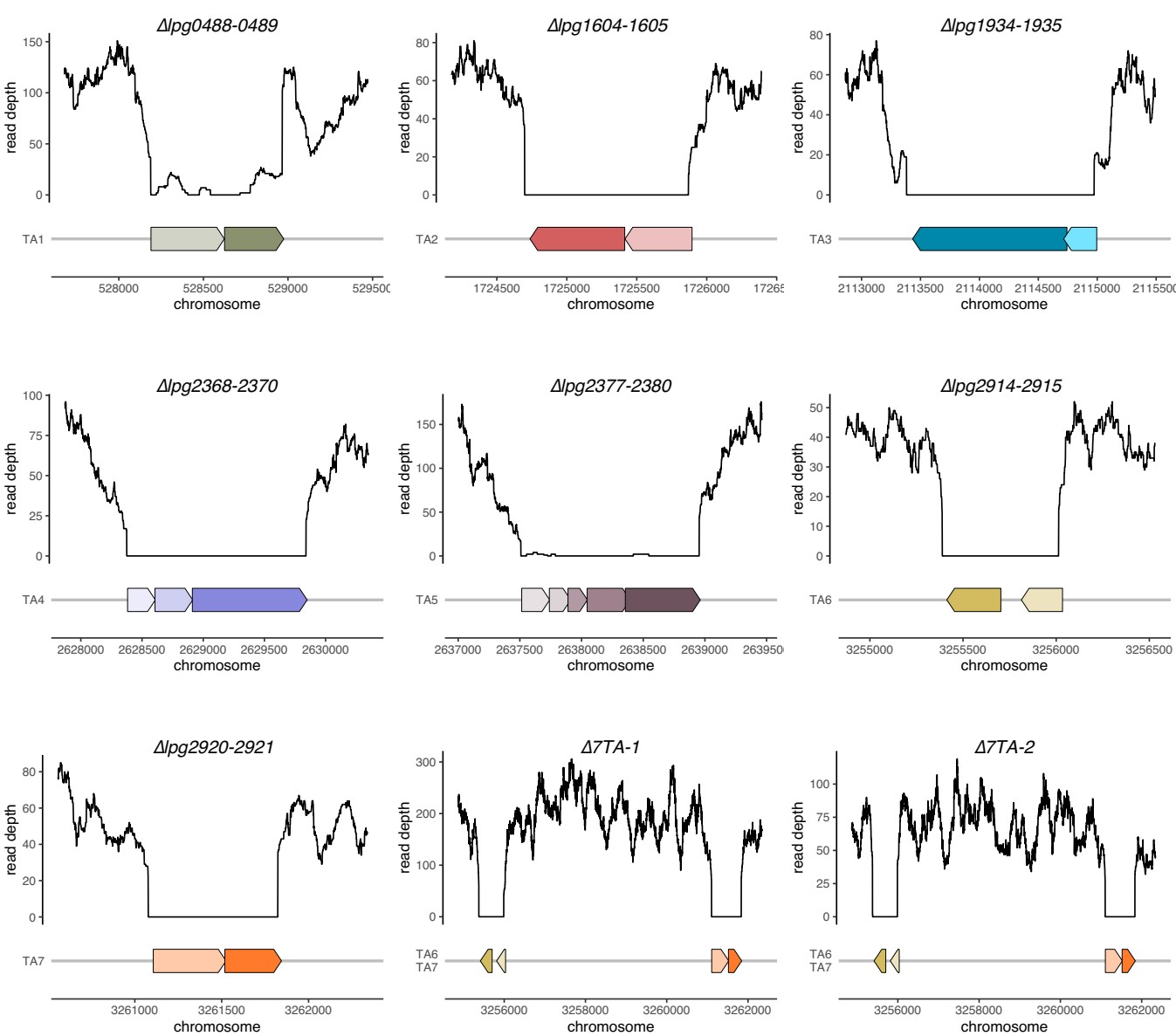

**Figure EV1. Genomic sequencing confirms TA system deletions.**

Sequencing read coverage of the *L. pneumophila* single and pan-TA deletion strains at the chromosomal locus of each TA system. Reads were aligned to a reference genome (Refseq ID: GCF_001941585.1) to confirm the fidelity of the edits. For the Δ7TA-1 and Δ7TA-2 strains, two TA loci are shown as an example. SNPs detected after sequencing are shown in Appendix Table S1.

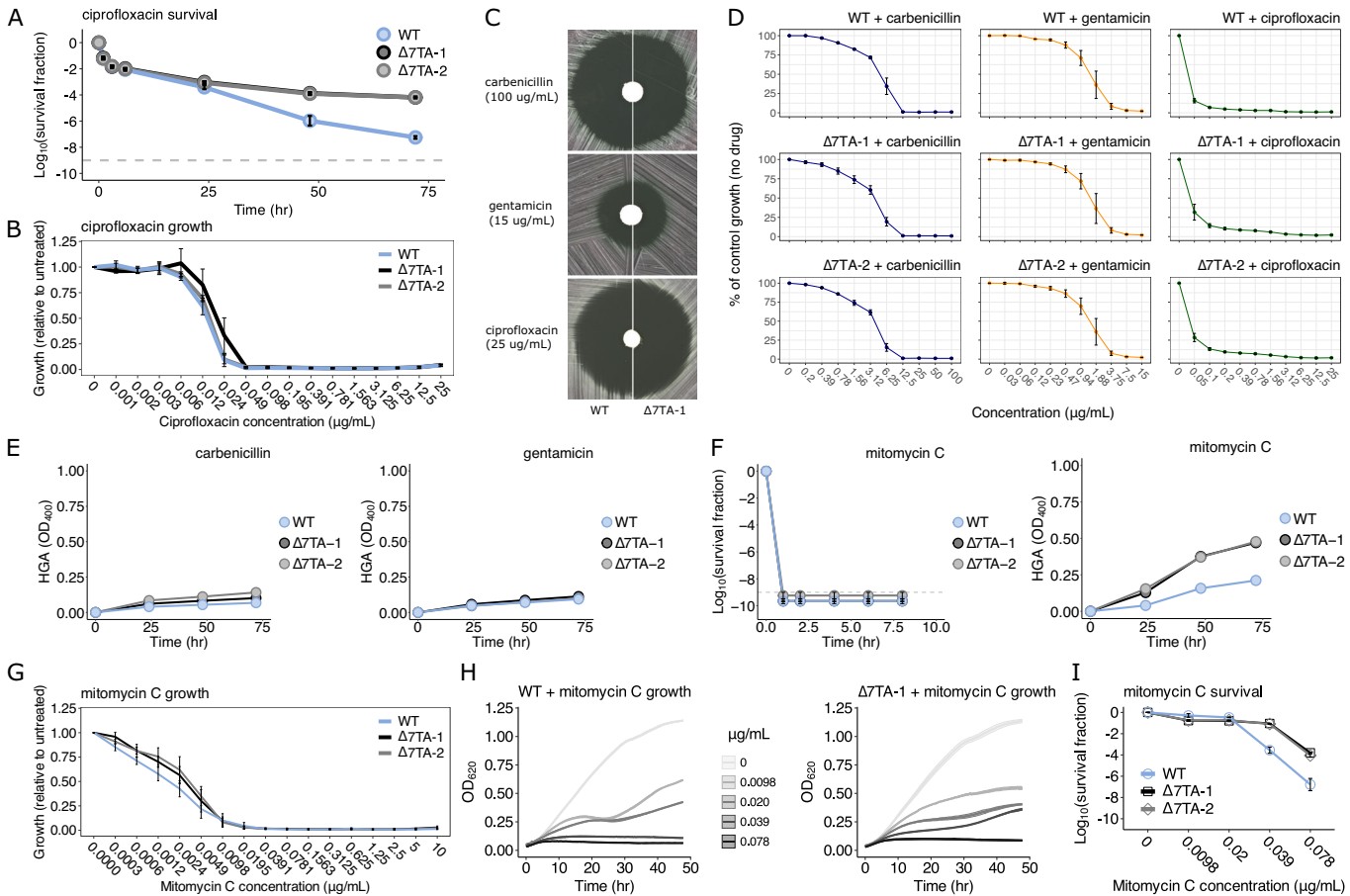

**Figure EV2. Antibiotic susceptibility is similar for the *L. pneumophila* wild-type and Δ7TA strains.**

(A) Time-kill assay measuring survival during treatment with ciprofloxacin ($n = 3$ biological replicates). (B) Ciprofloxacin MIC measurements for the wild-type, Δ7TA-1 and Δ7TA-2 strains ($n = 4$ biological replicates). (C) Disk diffusion assays for wild-type *L. pneumophila* and the Δ7TA-1 strain treated with carbenicillin, gentamicin, and ciprofloxacin (data are representative of $n = 2$ biological replicates). (D) Dose-response curves for wild-type (WT) *L. pneumophila* and two lineages of the Δ7TA strain treated with the antibiotics carbenicillin, gentamicin, and ciprofloxacin ($n = 2$ biological replicates). (E) HGA production by the wild-type and Δ7TA strains during treatment with carbenicillin and gentamicin (representative experiments are shown; $n = 2$ biological replicates). (F) Time-kill assay (left) and HGA quantification (right) of the wild-type and Δ7TA strains during treatment with mitomycin C. Survival data are from $n = 2$ biological replicates and HGA production from a representative experiment is shown ($n = 2$ biological replicates). (G) Mitomycin C MIC measurements for the wild-type, Δ7TA-1 and Δ7TA-2 strains ($n = 4$ biological replicates). (H) Dose-response curves for wild-type *L. pneumophila* and the Δ7TA-1 strain treated with mitomycin C at the indicated concentrations ($n = 2$ biological replicates). (I) Survival is shown after 48 h treatment with mitomycin C at the indicated concentrations for the wild-type, Δ7TA-1 and Δ7TA-2 strains ($n = 2$ biological replicates). Data information: In (A, B, D, F–I), data are presented as the mean (averaged for clarity) ± SEM. The limit of detection on all applicable plots is indicated with a dashed gray line.

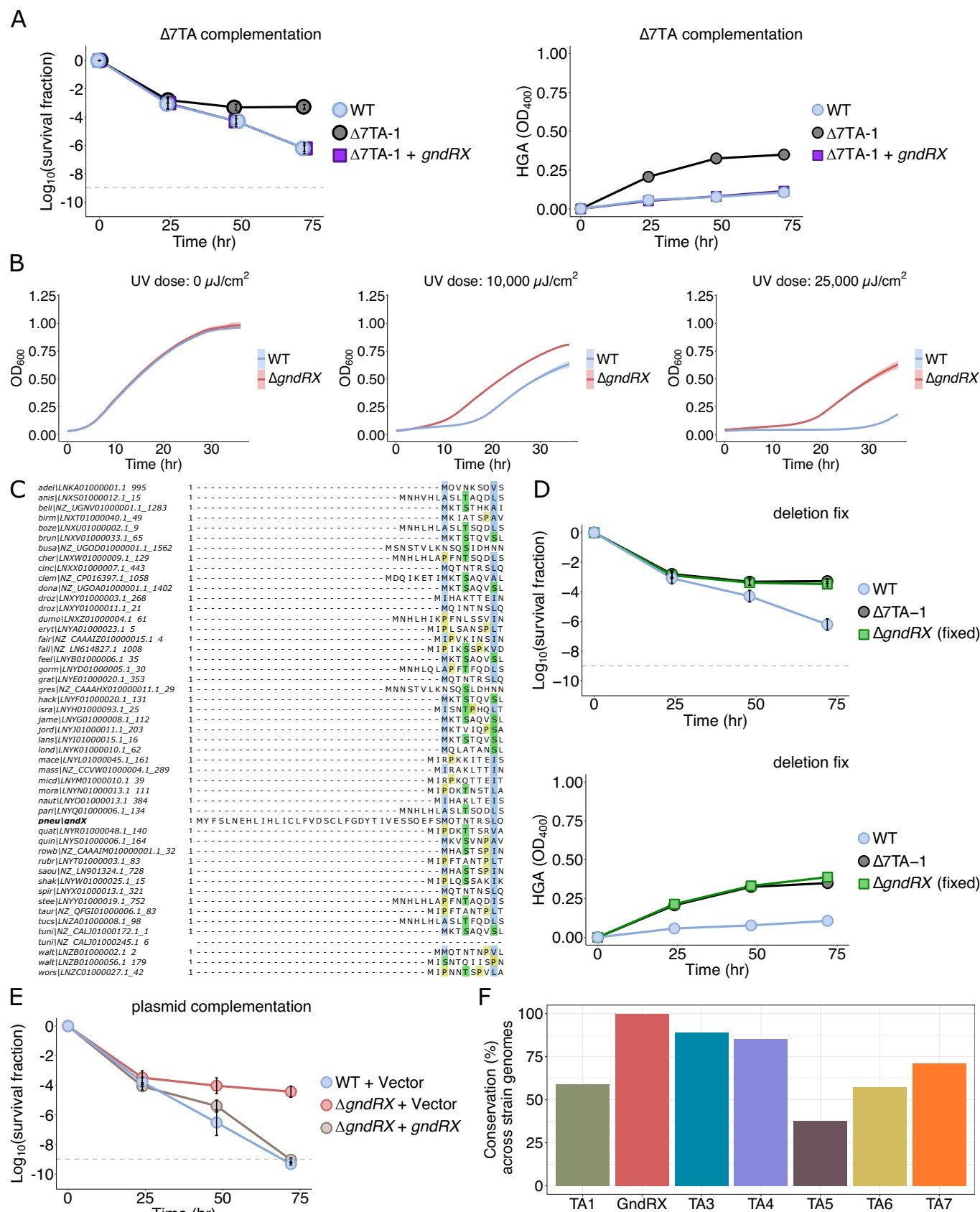

**Figure EV3. The *gndRX* locus is responsible for the genotoxic stress response phenotype.**

(A) Ciprofloxacin time-kill assay (left) and HGA quantification (right) to test the chromosomal complementation of the *gndRX* locus in the Δ7TA background. The Δ*gndRX* chromosomal lesion was repaired with a linear PCR product to restore the wild-type *gndRX* sequence and confirmed with Sanger sequencing. Survival data are from $n = 2$ biological replicates and HGA production from a representative experiment is shown ($n = 2$ biological replicates). (B) Growth curves of wild-type and Δ*gndRX* strains following UV irradiation at the indicated doses (data are representative of $n = 3$ biological replicates). (C) Multiple sequence alignment (MUSCLE) of GndX homologs found in *Legionella* species. *L. pneumophila* GndX is shown in bold. Residues conserved in at least 10% of proteins are colored with the Clustal X color scheme as implemented in Jalview. (D) Ciprofloxacin time-kill assay (top) and HGA quantification (bottom) for the wild-type, Δ7TA-1, and Δ7TA-1 (fixed Δ*gndRX* deletion) strains. The Δ*gndRX* chromosomal lesion in the Δ7TA-1 strain was repaired with a linear PCR product to restore the upstream 105 bp sequence that was misannotated as part of the *gndX* gene. The edit was confirmed with Sanger sequencing. Survival data are from $n = 2$ biological replicates and HGA production from a representative experiment is shown ($n = 2$ biological replicates). (E) Time-kill assay for plasmid complementation of *gndRX* in the Δ*gndRX* background. *L. pneumophila* strains carrying a plasmid (pNT562) with the *gndRX* sequence (without the misannotated 105 bp upstream region) or an empty vector control were treated with ciprofloxacin in the presence of kanamycin for plasmid maintenance. Survival data are from $n = 2$ biological replicates. (F) Bar chart of TA system conservation across 117 *L. pneumophila* strains with complete genomes in the NCBI Refseq database (Dataset EV1). Data information: In (A, B, D, E), data are presented as the mean (averaged for clarity) ± SEM. The limit of detection on all applicable plots is indicated with a dashed gray line.

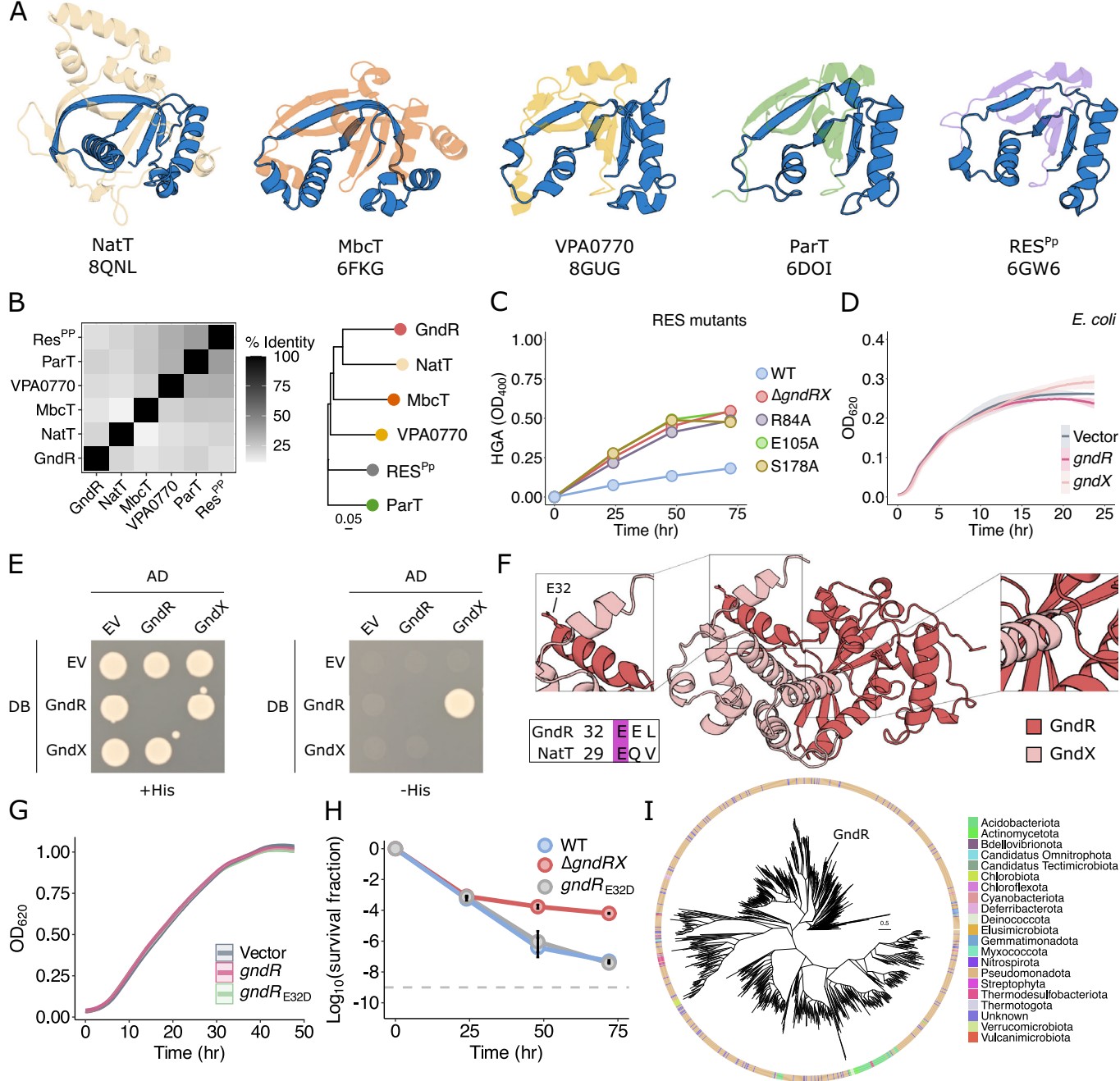

**Figure EV4. GndRX has remote homology to RES-Xre TA systems but non-canonical activity.**

(A) Crystal structures of NatT, MbcT, VPA0770, ParT, and RES[Pp]. The predicted RES domains in each protein are colored in blue. (B) Multiple sequence alignment (MUSCLE) similarity matrix and gene tree of GndR with the NatT, MbcT, VPA0770, ParT, and RES[Pp] toxins. (C) HGA production from *L. pneumophila* strains containing substitutions for individual R-E-S residues and treated with ciprofloxacin (a representative experiment is shown; $n = 3$ biological replicates). (D) Growth curves of *E. coli* expressing *gndR* or *gndX* compared with an empty vector (EV; pBAD33) control ($n = 2$ biological replicates). Expression was induced with 0.2% arabinose. (E) Yeast two-hybrid assay of GndRX protein-protein interactions (data are representative of $n = 2$ biological replicates). Constructs cloned with fusions to the activating domain (AD) or DNA-binding domain (DB) of the GAL4 transcription factor are indicated. Control experiments were performed in the presence of histidine (+His) and interaction experiments were performed in the absence of histidine (-His). (F) AlphaFold2 predicted interaction complex of GndR-GndX. Insets show the protein regions involved in the interaction (left) and putative regulatory site (right). A multiple sequence alignment between GndR and NatT is shown to highlight the conserved E32 residue in GndR (also shown in left inset). (G) Growth curves of *L. pneumophila* expressing *gndR*, *gndR*$_{E32D}$, or an empty vector control (pNT562). Expression was induced with IPTG (100 μM). Data are from $n = 3$ biological replicates. (H) Ciprofloxacin time-kill assay comparing wild-type, Δ*gndRX*, or a strain with the chromosomal mutation *gndR*$_{E32D}$. Survival data are from $n = 3$ biological replicates. (I) Phylogeny of GndR protein homologs detected in the UniProtKB database (Dataset EV2). Sequences were clustered at 80% similarity using MMseqs2 and a representative sequence was used for each cluster. Taxonomic assignment was performed with MMseqs2 using the lowest common ancestor (LCA) protocol. For each representative sequence, the LCA phylum is displayed as the color of the surrounding ring. The scale bars in both trees denotes substitutions per site. Data information: In (D, G, H), data are presented as the mean (averaged for clarity) ± SEM. The limit of detection on all applicable plots is indicated with a dashed gray line.

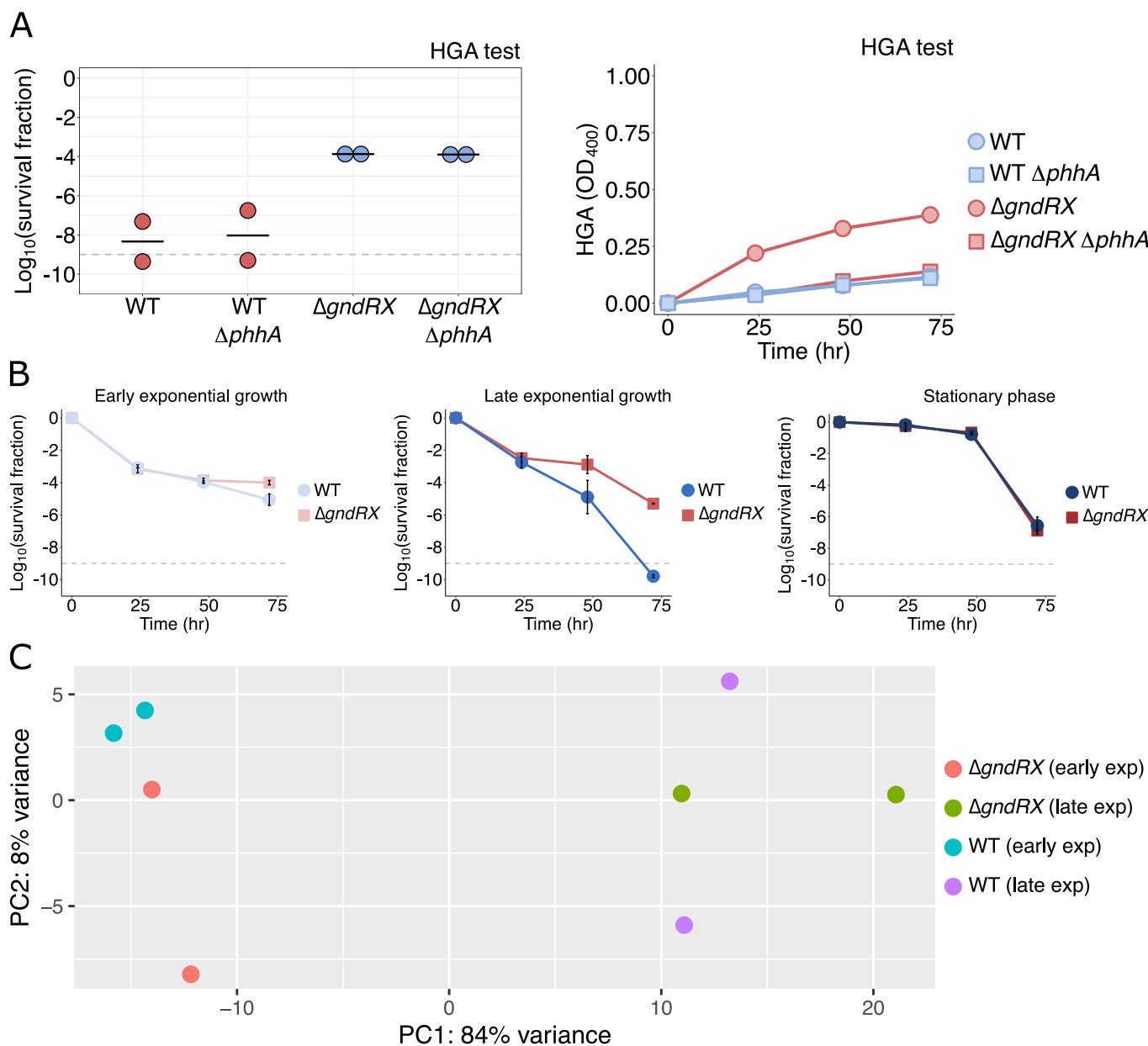

**Figure EV5. The stress survival phenotype is HGA-independent but growth phase-dependent.**

(A) Ciprofloxacin time-kill assay (left) and HGA quantification (right) of wild-type and Δ*gndRX* strains comparing the effect of deleting *phhA* and thereby abrogating HGA biosynthesis. The chromosomal *phhA* locus was deleted using the MazF recombineering technique and confirmed with Sanger sequencing. Survival data are from 72 h post-ciprofloxacin treatment ($n = 2$ biological replicates) and a representative experiment is shown for HGA quantification. (B) Time-kill assays comparing wild-type and Δ*gndRX* strains treated with ciprofloxacin at different growth phases. Cultures were grown overnight to either early (OD₆₀₀ = 1), late (OD₆₀₀ = 3), and stationary (OD₆₀₀ = ~4) growth phases prior to drug treatment. Data are from $n = 2$ biological replicates. (C) Principal coordinate plot comparing transcriptomes of wild-type and Δ*gndRX* strains grown to early and late exponential phase prior to RNA extraction and sequencing ($n = 2$ biological replicates for each strain + growth phase combination). Data information: In (A, B), data are presented as the mean (averaged for clarity) ± SEM. The limit of detection on all applicable plots is indicated with a dashed gray line.

