## [Peer Review File · EMBO Reports]

A bacterial toxin-antitoxin system involved in an unusual response to genotoxic stress

Jordan Lin, Beth Nicholson, and Alexander Ensminger

Corresponding author(s): Alexander Ensminger (alex.ensminger@utoronto.ca)

Review Timeline:

Submission Date:	18th Dec 24
Editorial Decision:	3rd Feb 25
Revision Received:	12th Jun 25
Editorial Decision:	9th Jul 25
Revision Received:	16th Jul 25
Accepted:	29th Jul 25

Editor: Achim Breiling

Transaction Report:

Dear Dr. Ensminger,

Thank you for the submission of your manuscript to EMBO reports. I have now received the reports from the three referees that were asked to evaluate your study, which can be found at the end of this email.

As you will see, the referees think that these findings are of interest. However, they have several comments, concerns, and suggestions, indicating that a major revision of the manuscript is necessary to allow publication of the study in EMBO reports. As the reports are below, and all the referee concerns need to be addressed, I will not detail them here.

Given the constructive referee comments, I would like to invite you to revise your manuscript with the understanding that the concerns of the referees must be addressed in the revised manuscript and in a detailed point-by-point response. Acceptance of your manuscript will depend on a positive outcome of a second round of review. It is EMBO reports policy to allow a single round of revision only and acceptance of the manuscript will therefore depend on the completeness of your responses included in the next, final version of the manuscript.

- 1) a .docx formatted version of the final manuscript text (including legends for main figures, EV figures and tables), but without the figures included. Figure legends should be compiled at the end of the manuscript text.
- 2) individual production quality figure files as .eps, .tif, .jpg (one file per figure), of main figures and EV figures. Please upload these as separate, individual files upon re-submission.

- 4) a complete author checklist, which you can download from our author guidelines (<https://www.embopress.org/page/journal/14693178/authorguide>). Please insert page numbers in the checklist to indicate where the requested information can be found in the manuscript. The completed author checklist will also be part of the RPF.

- 5) that primary datasets produced in this study (e.g. RNA-seq, ChIP-seq, structural and array data) are deposited in an

appropriate public database. If no primary datasets have been deposited, please also state this in a dedicated section (e.g. 'No primary datasets have been generated and deposited'), see below.

The accession numbers and database should be listed in a formal "Data Availability" section that follows the model below. This is now mandatory (like the COI statement). Please note that the Data Availability Section is restricted to new primary data that are part of this study. This section is mandatory. As indicated above, if no primary datasets have been deposited, please state this in this section

Data availability

8) Regarding data quantification and statistics, please make sure that the number "n" for how many independent experiments were performed, their nature (biological versus technical replicates), the bars and error bars (e.g. SEM, SD) and the test used to calculate p-values is indicated in the respective figure legends (also for EV and Appendix figures). Please also check that all the p-values are explained in the legend, and that these fit to those shown in the figure. Please provide statistical testing where applicable. Please avoid the phrase 'independent experiment', but clearly state if these were biological or technical replicates. Please also indicate (e.g. with n.s.) if testing was performed, but the differences are not significant. In case n=2, please show the data as separate datapoints without error bars and statistics. See also: <http://www.embopress.org/page/journal/14693178/authorguide#statisticalanalysis>

9) Please add scale bars of similar style and thickness to microscopic images, using clearly visible black or white bars (depending on the background). Please place these in the lower right corner of the images themselves. Please do not write on or near the bars in the image but define the size in the respective figure legend.

10) Please also note our reference format:

12) We now use CRediT to specify the contributions of each author in the journal submission system. CRediT replaces the author contribution section. Please use the free text box to provide more detailed descriptions and do NOT provide your final manuscript text file with an author contributions section. See also our guide to authors: <https://www.embopress.org/page/journal/14693178/authorguide#authorshipguidelines>

13) All Materials and Methods need to be described in the main text using our 'Structured Methods' format, which is required for

all research articles. According to this format, the Methods section should include a Reagents and Tools Table (listing key reagents, experimental models, software, and relevant equipment and including their sources and relevant identifiers), uploaded as separate file, and a Methods section in which we encourage the authors to describe their methods using a step-by-step protocol format with bullet points, to facilitate the adoption of the methodologies across labs. More information on how to adhere to this format as well as downloadable templates (.doc) for the Reagents and Tools Table can be found in our author guidelines (section 'Structured Methods'):

14) Please order the sections like this, using these names:

Title page - Abstract - Keywords - Introduction - Results - Discussion - Methods - Data availability section - Acknowledgements (including the funding information) - Disclosure and Competing Interests Statement - References - Figure legends - Expanded View Figure legends

15) Please make sure that all the funding information is also entered into the online submission system and that it is complete and similar to the one in the acknowledgement section of the manuscript text file.

I look forward to seeing a revised form of your manuscript when it is ready.

Yours sincerely,

Referee #1:

The ms by Lin & Ensminger describes the characterization of a particular toxin-antitoxin system (TA) in *L. pneumophila* that, upon genotoxic stress, trigger cell death. Under co-culture conditions, the TA-mutant cells enhance the survival of the wild-type cells through a mechanism that remains to be elucidated and requires cell-to-cell contact.

While I found the ms potentially interesting, there is a substantial number of questions and issues that must be resolved. Importantly, the GndRX system is homologous to the *Pseudomonas aeruginosa* NatRT system (Santi et al., EMBO J, 2004) (this information appears only in the discussion). Both systems share several characteristics: they are part of the core genome; the toxins are not toxic by themselves, and their activity is similar (depletion of NAD). I believe that this should be clearly stated early on in the introduction. The NatR antitoxin and NatT toxin should be included in the sequence and structure comparisons showed in figures 4 and S4.

The Jenal group showed that the NatR antitoxin appears to activate the NatT toxin, at specific levels. This should be tested with the GndRX system. Additionally, the construction of the gain-of-function mutation, NatTE29D, in the GndR toxin, assuming the conservation of the amino acid, and subsequent confirmation of the resulting phenotype, are recommended.

Figure 2. If I understand correctly, the authors are using 25 microgr/mL of CIP, which seems to be above the MIC (matching the guidelines to study persistence) - although the MIC of the 2 strains are not reported but can be inferred from figure S2. Classical MIC measurements should be performed.

Although the authors use the term 'persistence', the time-kill curve for the WT strain does not show a classical biphasic curve (the rate of killing seems to be constant). On the contrary, the delta TA mutant shows a biphasic curve, indicative of persistence. In figure 7, survival is measured at shorter time-points (up to 25h). In this graph, the curves are biphasic, and the survival is similar for the 2 strains. This is quite confusing and should be carefully interpreted. A possible interpretation is that the deletion of the TA system enhance survival during long-term treatment with CIP. The authors should provide the complete time-kill curve with appropriate timepoints; all along the duration of the experiments (Figure 2F and 7B bare difficult to compare due to different

Y scales).

Figure S2D. The authors tested another genotoxic. Mitomycin C appears to be significantly more potent than ciprofloxacin (time scale for killing the entire population is far shorter than with ciprofloxacin). No difference is detected in terms of survival between the 2 strains. How much is the MIC for mitomycin C? Is it similar for the 2 strains? In Figure S2E, the authors measure the OD at varying doses, however they should measure survival. I don't think they can conclude that there are dose-dependent differences in susceptibility only based on OD measurements.

Importantly, how the authors reconcile the phenotype observed under ciprofloxacin and mitomycin C treatment? The HGA production appears to be higher for the WT strain in mitomycin C than in ciprofloxacin. Do the authors have some explanation?

The RNA-Seq data should be presented differently. For instance, in Figure 5, more 'dots' should be assigned to gene names, especially those are the most extreme. More useful information can certainly be extracted and presented.

Figure S5. The graphs presented in the B panel should be overlaid. It is quite difficult to see the differences between the 2 strains. How do the authors explain the absence of difference when cells are treated in stationary phase? It seems that there is not much difference in the 'early' condition either. Possibly related to that is the data regarding the 9:1 ratio between the WT and the mutant, where the mutant 'loses' the capacity to survive better than the WT. Is it a question of cell density? This hypothesis should be tested.

Figure 7A and figure S7A. The OD are quite different between the 2 experiments. Could the authors comment on that?

Figure 7B. The survival is similar for both strains. However, the WT is lysing and about half of the population is PI-stained, while only 2% of the mutant cells are PI stained. It would be helpful to show the non-treated conditions to have a better sense of the gating used. Also, the LacZ test revealed that the mutant is lysing as well. However the OD remains constant. How to reconcile these observations? That OD measurement is a poor proxy?

Figure 7D. The intracellular concentration of NAD drastically drops in the WT cells, suggesting that they are metabolically inactive ('dormant'). However, the authors claim in the discussion (L483) that the mutant cells are in a dormant state that helps them to survive. It's a bit confusing. What is the definition of dormancy and the evidence supporting the hypothesis that dormancy helps to survive?

On an evolutionary point of view, what would be the advantage of selecting a TA system that is deleterious, integrating it in the host regulatory network and maintaining it? This system seems to be widespread in bacteria. This should be discussed.

Minor comments:

L62. Add references 44 and 45.

L105. Why is the repertoire 'reduced'? Not all the bacterial strains have 'a lot' of TA systems. E.g. The E. coli lab strain has 10 identified type II TA systems.

The phylogenetic tree presented in Figure S4 is unreadable.

Referee #2:

The manuscript describes the role of Legionella toxin-antitoxin systems in regulation of cell physiology. The experiments indicate unexpected effects: deletion of one of the TA systems decreases sensitivity to ciprofloxacin; the effect is regulated by cell to cell contact. The results are of high interest for the colleagues in the field and are also important for wider audience. The experiments are well planned and performed, the manuscript reads well.

There is one piece of experimental evidence that I find missing: what happens to NAD levels if you overexpress *gndR*? It is speculated in the manuscript that overexpression of *gndR* might influence NAD levels to the extent that does not effect growth. This would be relatively easy experiment to perform, considering that the authors have the experimental system set up.

As a very minor note, symbols are not visible in Fig 3B left. Please find a more clear graphical solution. In addition, the Fig.3B left has smaller variability of data when compared to the other datasets. Was the same number of replicates performed correctly?

I find the Discussion lengthy. The authors might consider shortening.

Referee #3:

The manuscript by Lin and Ensminger investigates toxin-antitoxin (TA) systems of *Legionella pneumophila*, the causative agent of Legionnaires' disease. The study documents the construction of a marker-less "pan-TA" *L. pneumophila* deletion strain lacking all 7 annotated TA systems and the characterization of stress responses of this mutant strain. Intriguingly, one of the 7 TAs, TA2 (Lpg1604-1605, GndRX), is activated specifically by genotoxic stress (ciprofloxacin, mitomycin C, UV) and promotes bacterial cell death rather than survival. The deletion of *gndRX* does not affect bacterial growth, host cell infection, or survival during translation and cell wall stress. However, a Δ *gndRX* mutant strain shows increased production of the metabolite homogentisic acid (HGA) and greatly enhanced survival (in a dormant state). These features are conferred to wild-type bacteria in a contact-dependent manner and are affected by the ratio of Δ *gndRX* mutant to wild-type bacteria. In contrast to canonical TA systems, the overproduction of GndR or GndX is not toxic for *L. pneumophila* or *E. coli*. GndR interacts with GndX (as shown by yeast two hybrid), likely binds NAD⁺ through its conserved RES (arginine, glutamine, serine) domain and promotes cellular NAD⁺ depletion upon the exposure to genotoxic stress. By promoting contact-dependent survival, GndRX reveals an interesting new function for bacterial TA systems.

The study represents an exciting analysis of a novel bacterial TA system with an unprecedented function. The thorough investigation comprises many appropriate controls, including an assessment of a "fixed" Δ *gndRX* mutant strain. The story unfolds in a straightforward, logical manner, and the well-written manuscript is actually a pleasure to read. A few minor amendments would further strengthen the study.

1) Fig. 4D: The asterisk mentioned in the figure legend (supposedly highlighting the conserved serine) is actually an arrow. Please fix.

2) Statistics: the calculation of the SEM should include at least 3 independent experiments. This is apparently not the case for Fig. 2DE, 6DE, 7B, S2ABD, S3ADE, S4DE, S5BC, and S7AB. Please show individual data points. Also, please indicate the overall number of experiments performed for Fig. 2G, S2C, S3A, and S5A.

3) Line 378 or Discussion section: please discuss "VBNC" *L. pneumophila* in more detail, e.g. PMID: 23968544, PMID: 39810465.

4) Line 445: possibly also cite PMID: 34314090.

We wish to thank the editor and reviewers for their thoughtful, constructive comments and careful consideration of our manuscript.

A point-by-point response to previous reviews is provided below:

Referee #1:

*The ms by Lin & Ensminger describes the characterization of a particular toxin-antitoxin system (TA) in *L. pneumophila* that, upon genotoxic stress, trigger cell death. Under co-culture conditions, the TA-mutant cells enhance the survival of the wild-type cells through a mechanism that remains to be elucidated and requires cell-to-cell contact.*

*While I found the ms potentially interesting, there is a substantial number of questions and issues that must be resolved. Importantly, the GndRX system is homologous to the *Pseudomonas aeruginosa* NatRT system (Santi et al., EMBO J, 2004) (this information appears only in the discussion). Both systems share several characteristics: they are part of the core genome; the toxins are not toxic by themselves, and their activity is similar (depletion of NAD). I believe that this should be clearly stated early on in the introduction. The NatR antitoxin and NatT toxin should be included in the sequence and structure comparisons showed in figures 4 and S4.*

We thank the reviewer for their careful reading of our manuscript and constructive feedback, which has helped us to revise our work with additional experiments and analyses. We feel this has resulted in a much improved and more complete story. We provide point-by-point responses below.

The NatRT paper was published shortly before our manuscript was submitted and thus, we did not provide a more in-depth comparison in our initial submission. Our revised manuscript now includes the NatRT system for both the sequence and structure comparisons in Figures 4 and S4, as well as functional comparisons between GndRX and NatRT (detailed in the next section). We show that among the RES-Xre TA systems, GndRX is most closely related to the NatRT system, including the N-terminal 'flap' motif in the toxin protein, and both systems share little sequence similarity with the other RES-Xre systems. However, despite their overall structural similarity GndRX and NatRT are quite divergent homologs and we propose that critical functional differences underlie the distinct physiological roles that these systems perform.

Finally, we now include in-depth comparisons of the NatRT and GndRX systems in both the results and discussion sections on pages 9, 10, and 20.

The Jenal group showed that the NatR antitoxin appears to activate the NatT toxin, at specific levels. This should be tested with the GndRX system. Additionally, the construction of the gain-of-function mutation, NatTE29D, in the GndR toxin, assuming the conservation of the amino acid, and subsequent confirmation of the resulting phenotype, are recommended.

We thank the reviewer for recommending these experiments, as they have yielded key insights that we feel distinguish the GndRX and NatRT systems in a way that reflects the underlying differences in the physiological roles of these systems. To address the reviewer's suggestions, we constructed a two-inducer co-expression system in *L. pneumophila* to compare the effect of varying the ratios of GndR and GndX on cell growth. This was done in the same manner as was used for NatR and NatT and showed only a minor effect on cell growth at the highest tested expression levels of GndR. These results are presented in a revised Figure 4.

Furthermore, the E29 residue is indeed conserved in GndR (E32) and we tested both ectopic expression and ciprofloxacin survival of the analogous mutation (E32D) in *L. pneumophila*. Surprisingly, in neither instance did we observe an effect on system function, suggesting that while the GndRX system is most closely related to NatRT, it has undergone additional functional divergence which supports our hypothesis of domestication of an ancestral TA system. We now present these experimental findings in revised Figures 4 and EV4.

Figure 2. If I understand correctly, the authors are using 25 microgr/mL of CIP, which seems to be above the MIC (matching the guidelines to study persistence) - although the MIC of the 2 strains are not reported but can be inferred from figure S2. Classical MIC measurements should be performed.

We have performed these experiments and provide the results in a new figure panel (Figure EV2B), as well as mentioning these values in the text on lines Page 6 (Lines 142-144):

“Consistent with this, we confirmed that the minimum inhibitory concentration (MIC) of ciprofloxacin was the same across strains (Figure EV2B) and that our treatment concentration (25 µg/mL) far exceeded the MIC (>100x).”

Although the authors use the term 'persistence', the time-kill curve for the WT strain does not show a classical biphasic curve (the rate of killing seems to be constant). On the contrary, the delta TA mutant shows a biphasic curve, indicative of persistence.

In figure 7, survival is measured at shorter time-points (up to 25h). In this graph, the curves are biphasic, and the survival is similar for the 2 strains. This is quite confusing and should be carefully interpreted. A possible interpretation is that the deletion of the TA system enhance survival during long-term treatment with CIP. The authors should provide the complete time-kill curve with appropriate timepoints; all along the duration of the experiments (Figure 2F and 7B bare difficult to compare due to different Y scales).

We thank the reviewer for these insightful observations and appreciate that the presentation of our time-kill assay data across two separate plots may have led to some confusion. Our interpretation is that both the WT and $\Delta 7TA$ strains exhibit a classical biphasic kill curve along the conventional timescale of several hours. Subsequent to this, and after prolonged incubation with antibiotics, the strains diverge in their survival/viability. We are curious as to whether this is due to differences in cellular resuscitation from dormancy, for example, however the exact mechanisms underlying this fascinating phenomenon remain unclear. As requested by the reviewer, we now include experimental data from time-kill assays performed with appropriate time points across the full duration of the experiment in a panel Figure EV2A.

Figure S2D. The authors tested another genotoxic. Mitomycin C appears to be significantly more potent than ciprofloxacin (time scale for killing the entire population is far shorter than with ciprofloxacin). No difference is detected in terms of survival between the 2 strains. How much is the MIC for mitomycin C? Is it similar for the 2 strains? In Figure S2E, the authors measure the OD at varying doses, however they should measure survival. I don't think they can conclude that there are dose-dependent differences in susceptibility only based on OD measurements. Importantly, how the authors reconcile the phenotype observed under ciprofloxacin and mitomycin C treatment? The HGA production appears to be higher for the WT strain in mitomycin C than in ciprofloxacin. Do the authors have some explanation?

We again thank the reviewer for the suggested experiments as we feel that the data from them has strengthened our manuscript. We now include MIC, broth growth, and survival measurements during mitomycin C (MMC) treatment for both the WT and $\Delta 7TA$ strains in revised Figure panels EV2G/H/I. These data demonstrate that while the MIC for MMC is the same for both strains, growth and survival at or above the MIC concentration (as measured by paired growth curves and CFU counts) is not. This suggests that these two strains may differ in how they tolerate DNA stress rather than their intrinsic susceptibility to it. We feel that this new dose-dependent survival data is much more definitive than the OD measurements we showed previously, as the reviewer noted.

Regarding the relative HGA production for the WT strain between ciprofloxacin and MMC, we do not have a strong explanation for why this is occurring. If we were to speculate, perhaps it is the consequence of the difference in killing kinetics between these two drugs and thus the severity of DNA stress being experienced by the WT during MMC treatment relative to ciprofloxacin. In the case of the $\Delta gndRX$ genotype, HGA production is associated with enhanced cell survival and reduced NAD depletion. It may be the case that WT cells treated with MMC adopt a predominantly dormant rather than dying state (as with ciprofloxacin) or undergo less NAD depletion or reactive oxygen species production, the outcome of which is increased HGA production. However, future work will be required to establish a clear mechanistic understanding of how DNA stress is connected to HGA production in these strains.

The RNA-Seq data should be presented differently. For instance, in Figure 5, more 'dots' should be assigned to gene names, especially those are the most extreme. More useful information can certainly be extracted and presented.

We have now included more gene labels for the dots with the strongest signal on the volcano plot in Figure 5. Unfortunately, the *L. pneumophila* genome is not well annotated and there is limited functional information at the individual gene level for many of the most differentially enriched genes. Thus in many cases we can only include gene identifiers, however these will be of use to those in the field that are familiar with *Legionella* genetics. Furthermore, we have added the following lines in the results section on Page 12 (lines 303-308):

“Many of the differentially enriched genes are uncharacterized however, such as *lpg1327*, which showed the largest fold change during genotoxic stress but has no predicted function. Interestingly, the gene upstream of *gndRX* (*lpg1603*) was also highly enriched in the wild type relative to the deletion strain, suggesting a possible transcriptional coupling between these genes. The protein product Lpg1603 is a predicted phospholipase but is otherwise uncharacterized, thus it remains unclear whether this protein has any functional relationship with GndRX.”

Figure S5. The graphs presented in the B panel should be overlaid. It is quite difficult to see the differences between the 2 strains. How do the authors explain the absence of difference when cells are treated in stationary phase? It seems that there is not much difference in the 'early' condition either. Possibly related to that is the data regarding the 9:1 ratio between the WT and the mutant, where the mutant 'loses' the capacity to survive better than the WT. Is it a question of cell density? This hypothesis should be tested.

We had initially chosen to display the differences within each strain across growth phases, however we appreciate that like the reviewer, other colleagues may be most interested in the comparison between strains at each growth phase. We have therefore adapted Figure EV5B to now display this data as suggested.

Our findings do indeed show the largest magnitude of survival difference between strains in mid-exponential growth phase and no difference in stationary phase. One explanation for this is that cells in stationary phase are typically highly tolerant to antibiotics due to their progression toward a dormant state and the bias of antibiotics toward dividing cells (Luidalepp et al. 2011, J Bacteriol). With regard to early exponential phase, one possibility is that cells at this time point may also not be rapidly dividing and thus the metabolic consequences of genotoxic stress are reduced relative to exponential growth. In our opinion, these growth phase-dependent differences in survival are likely distinct from the density-dependent survival differences we observed when altering the ratio of WT: $\Delta gndRX$ cells, which may instead be the consequence of the dilution of some protective factor of $\Delta gndRX$ cells, such as a membrane protein. In summary, cells that are rapidly dividing are the most susceptible to antibiotics and we speculate that the differences in survival between the WT and $\Delta gndRX$ strains are most apparent at this time of maximum damage.

Figure 7A and figure S7A. The OD are quite different between the 2 experiments. Could the authors comment on that?

These differences are due to the path length differences of a cuvette and 96-well plate when performing spectrophotometric measurements. To assist others in the interpretation of these data, we have included a note in each figure legend indicating this.

Figure 7B. The survival is similar for both strains. However, the WT is lysing and about half of the population is PI-stained, while only 2% of the mutant cells are PI stained. It would be helpful to show the non-treated conditions to have a better sense of the gating used. Also, the LacZ test revealed that the mutant is lysing as well. How the OD remains constant. How to reconcile these observations? That OD measurement is a poor proxy?

The reviewer makes excellent observations, and we agree that these orthogonal readouts are complicated to interpret, which is why we chose to include all of them to complement each of their respective limitations. We now include a revised supplementary Figure S2C which shows the untreated controls (live untreated and heat-killed cells) used in the gating for Figure 7C. Furthermore, our interpretation is indeed that absorbance is a coarse measurement in this instance, as it can only capture cell death dynamics if lysis occurs. Thus we chose to use turbidity to measure cell replication kinetics, the LacZ to measure lysis, flow cytometry to monitor the state of cell damage, and CFUs to measure the persister subpopulation. Our interpretation is that many $\Delta gndRX$ cells in the bulk population continue to divide (or at least attempt to) during the first 24h of drug treatment (Figure 7A shows an increase in turbidity for this strain). We hypothesize that this is the consequence of not having the GndRX system and thus maintaining higher levels of NAD after stress exposure. However, these cells are still being incubated with large amounts of drug and thus during this time a subset of cells also undergo lysis. These concurrent phenomena result in the OD appearing to remain relatively constant while also producing a signal in the LacZ assay. During this time, the persister subfraction (which is a small number of cells relative to the bulk population) declines equivalently for each strain as these are likely quasi-dormant cells that are unaffected by the death kinetics of the broader population.

Figure 7D. The intracellular concentration of NAD drastically drops in the WT cells, suggesting that they are metabolically inactive ('dormant'). However, the authors claim in the discussion (L483) that the mutant cells are in a dormant state that helps them to survive. It's a bit

confusing. What is the definition of dormancy and the evidence supporting the hypothesis that dormancy helps to survive?

We appreciate this feedback from the reviewer and have refined the writing of our manuscript to be more specific when describing cells as dormant. The term “dormancy” is complex and nebulous, and likely encompasses many different cell states along a continuum of growth arrest (Walker et al. 2024, Trends Microbiol). While there currently isn’t a consensus in the field on what dormancy specifically describes (McDonald et al. 2024, Trends Microbiol), we generally view it as a cell state of highly reduced metabolism and a reversible cessation of growth and replication. The adoption of a dormant or non-replicative state can enhance cell survival to diverse stresses (Lennon et al. 2011, Nat Rev Microbiol.), such as antibiotics, due in part to the targeting of replicating cells. In our experimental system, we observed that reduction in cellular NAD is coincident with cell death. This is consistent with the activity of the other RES toxins, which deplete or consume NAD, and as a direct or indirect consequence, this leads to cell death. Thus we consider NAD depletion to be a property of dying cells rather than dormant cells, which likely require the retention of some level of critical factors (like NAD) to remain viable even if they are not dividing. We refer to cells in the mutant population as dormant because they are not dead or dying, retain higher levels of NAD, and likely represent a characterized state of dormancy referred to as the viable but non-culturable (VBNC) state. VBNC cells have been studied in *L. pneumophila* previously, including with the methodologies we use to detect these populations (flow cytometry and CFU quantification). As dormancy is indeed a complex term associated with many different cell states, we have modified our text to refer specifically to the VBNC state.

On an evolutionary point of view, what would be the advantage of selecting a TA system that is deleterious, integrating it in the host regulatory network and maintaining it? This system seems to be widespread in bacteria. This should be discussed.

We thank the reviewer for this insightful question and agree that the presence of this system across bacterial clades seems counterintuitive. Our hypothesis is that this system could have selective benefits during stress conditions (such as phage infection or general genotoxic stress) whereby total population fitness is enhanced by the eradication of infected or irreversibly damaged cells. This could allow for healthy kin cells to access limiting nutrients or prevent the spread of infectious agents and deleterious mutations in the population. Conversely, this system may also provide a means for growth arrest to facilitate repair—rather than lead to outright cell death—under physiological stress conditions rather than the extreme stresses of the laboratory environment. Given the fascinating and seemingly paradoxical conservation of this system across bacteria, we speculate about its evolutionary success in the discussion starting on page 21 (lines 546-563):

“These observations therefore raise the question: why have this system in the first place? The rapidly expanding catalogue of bacterial immune systems has revealed both the prevalence of NAD⁺ depletion as a mode of defense (Boyle & Hatoum-Aslan, 2023) and similar instances of TA-based systems that rely on both proteins for toxic activity (Burman et al, 2024). Perhaps, GndRX may function as one component of a larger response pathway and serves an intermediary role in converting a DNA damage signal into the depletion of a critical cellular metabolite, thereby acting as a homeostatic sensor rather than an autonomous immune element. Furthermore, it is possible that cell death is in fact an exacerbated consequence of the magnitude and duration of genotoxic stress exposure we utilized, and this system would otherwise produce a bacteriostatic effect under physiological conditions. For example, when experiencing acute DNA stress, it may be advantageous for cells to restrict growth to reduce

oxidative damage and await improving conditions for repair to proceed. Alternatively, as not all wild-type cells die even after prolonged damage, activation of GndRX may instead be a means of heterogeneity generation and cellular altruism. As vacuolar compartments and biofilm communities are inherently spatially constrained environments with finite resources, it could be advantageous for damaged or genotypically compromised cells to undergo growth restriction or programmed cell death to liberate replicative resources for kin cells. Finally, it is possible that GndRX is indirectly activated upon DNA stress induction and performs a secondary function, which under conditions of prolonged and irreparable genotoxic stress, leads to cell death.”

Minor comments:

L62. Add references 44 and 45.

Done.

L105. Why is the repertoire 'reduced'? Not all the bacterial strains have 'a lot' of TA systems. E.g. The *E. coli* lab strain has 10 identified type II TA systems.

We used the term reduced because we were referring specifically to genetically tractable bacteria studied in the laboratory and not the overall distribution of TA systems across bacteria. *L. pneumophila* has a relatively small set of TA systems compared to typical models (*E. coli*, *Salmonella*, *Mtb*) in that it has both fewer type II systems and no predicted systems of other types. So compared with these systems, in which the majority of TA research has been conducted, the total number of TA systems in *L. pneumophila* is much fewer than the total number in these other model bacteria. For clarity, we have reworded “reduced” to “limited”.

The phylogenetic tree presented in Figure S4 is unreadable.

We have simplified the phylogeny in Figure EV4 by excluding taxonomic information at the Class level and now only show the Phylum level. Our intention is only to convey the broad distribution of GndR homologs across bacterial phyla rather than specific details about taxonomic associations.

Referee #2:

The manuscript describes the role of Legionella toxin-antitoxin systems in regulation of cell physiology. The experiments indicate unexpected effects: deletion of one of the TA systems decreases sensitivity to ciprofloxacin; the effect is regulated by cell to cell contact. The results are of high interest for the colleagues in the field and are also important for wider audience. The experiments are well planned and performed, the manuscript reads well.

There is one piece of experimental evidence that I find missing: what happens to NAD levels if you overexpress gndR? It is speculated in the manuscript that overexpression of gndR might influence NAD levels to the extent that does not effect growth. This would be relatively easy experiment to perform, considering that the authors have the experimental system set up.

We thank the reviewer for their positive feedback on our manuscript and this very helpful experimental suggestion. We have performed the experiments as requested and now include these data in a revised Figure 7E. We show that expression of GndR and GndR+GndX both result in the depletion of cellular NAD, though not to levels seen during even the 6h of genotoxic

stress in WT cells. Additionally, we compared NAD depletion across both exponential and post-exponential growth phases to confirm that the strains grew normally and could reach saturating growth. These findings therefore support the hypothesis that both GndR and GndR+GndRX can deplete cellular NAD when expressed, and that this activity does not cause sufficient toxicity to the cell such that growth is arrested or death occurs.

As a very minor note, symbols are not visible in Fig 3B left. Please find a more clear graphical solution. In addition, the Fig.3B left has smaller variability of data when compared to the other datasets. Was the same number of replicates performed correctly?

We have switched and dispersed the symbols in Figure 3B left to make this figure more accessible to the reader. The same number of independent experiments (ie biological replicates) was performed to generate the data in Fig 3B as with other experiments, however in this instance the variation was very small between the $\Delta gndRX$, $\Delta gndR$, and $\Delta gndX$ strains which resulted in unavoidable overlap between the data points on the plot.

I find the Discussion lengthy. The authors might consider shortening.

We thank the reviewer for this feedback and have edited the discussion section for brevity while retaining all the core points.

Referee #3:

The manuscript by Lin and Ensminger investigates toxin-antitoxin (TA) systems of Legionella pneumophila, the causative agent of Legionnaires' disease. The study documents the construction of a marker-less "pan-TA" L. pneumophila deletion strain lacking all 7 annotated TA systems and the characterization of stress responses of this mutant strain. Intriguingly, one of the 7 TAs, TA2 (Lpg1604-1605, GndRX), is activated specifically by genotoxic stress (ciprofloxacin, mitomycin C, UV) and promotes bacterial cell death rather than survival. The deletion of gndRX does not affect bacterial growth, host cell infection, or survival during translation and cell wall stress. However, a $\Delta gndRX$ mutant strain shows increased production of the metabolite homogentisic acid (HGA) and greatly enhanced survival (in a dormant state). These features are conferred to wild-type bacteria in a contact-dependent manner and are affected by the ratio of $\Delta gndRX$ mutant to wild-type bacteria. In contrast to canonical TA systems, the overproduction of GndR or GndX is not toxic for L. pneumophila or E. coli. GndR interacts with GndX (as shown by yeast two hybrid), likely binds NAD⁺ through its conserved RES (arginine, glutamine, serine) domain and promotes cellular NAD⁺ depletion upon the exposure to genotoxic stress. By promoting contact-dependent survival, GndRX reveals an interesting new function for bacterial TA systems.

The study represents an exciting analysis of a novel bacterial TA system with an unprecedented function. The thorough investigation comprises many appropriate controls, including an assessment of a "fixed" $\Delta gndRX$ mutant strain. The story unfolds in a straightforward, logical manner, and the well-written manuscript is actually a pleasure to read. A few minor amendments would further strengthen the study.

We thank the reviewer for their thorough reading and positive assessment of our manuscript.

1) Fig. 4D: The asterisk mentioned in the figure legend (supposedly highlighting the conserved serine) is actually an arrow. Please fix.

We thank the reviewer for noticing this and have changed the asterisk to an arrow.

2) *Statistics: the calculation of the SEM should include at least 3 independent experiments. This is apparently not the case for Fig. 2DE, 6DE, 7B, S2ABD, S3ADE, S4DE, S5BC, and S7AB. Please show individual data points. Also, please indicate the overall number of experiments performed for Fig. 2G, S2C, S3A, and S5A.*

We have made the requested changes by adding the number of experiments performed to all figure captions and we now include individual data points for all of the main figures except for Figure 7B, which was made too difficult to read when all replicate data were included. This was similarly the case for the figures in the supplement, thus for the purposes of readability and not expanding many of the already large figures with more panels, we have retained the summary statistics in the supplemental plots. However, we have provided all the raw data values for the main figures in the Source Data Files.

3) *Line 378 or Discussion section: please discuss "VBNC" *L. pneumophila* in more detail, e.g. PMID: 23968544, PMID: 39810465.*

We have added sections on the VBNC state in *L. pneumophila* to the results and discussion:

(page 16, lines 410-411)

"In this state, *L. pneumophila* VBNC cells have been shown to retain membrane potential and metabolic activity but are unable to be resuscitated using conventional culturing (Schmid & Hilbi, 2025)."

(page 16, lines 415-418)

"Given that *L. pneumophila* VBNC cells are characterized by distinct transformations in their ultrastructure (Al-Bana et al, 2014), an important goal of future work will be to determine whether $\Delta gndRX$ cells adopt similar morphological changes."

(page 18, lines 460-464)

"Within *Legionella* biology, VBNC cells are a common developmental form adopted in response to diverse environmental stresses and are critical to the bacterium's capacity for prolonged survival during harsh extracellular conditions (Robertson et al, 2014). Thus, by regulating the transition to either death or the VBNC state, GndRX appears to play an important role in the bacterium's life cycle as opposed to mere selfish parasitism."

4) *Line 445: possibly also cite PMID: 34314090.*

Done.

Dear Prof. Ensminger

Thank you for the submission of your revised manuscript. I have now received the reports from the two referees that I asked to re-evaluate the study, you will find below. As you will see, both referees now support the publication of your study.

Before we can proceed with formal acceptance, I have these editorial requests I ask you to address in a final revised manuscript:

- Please provide the abstract written in present tense throughout.
- Please order the manuscript sections like this, using (only) these names:
Title page - Abstract - Keywords - Introduction - Results - Discussion - Methods - Data availability section - Acknowledgements - Disclosure and Competing Interests Statement - References - Figure legends - Expanded View Figure legends
- Please remove now the referee tokens from the data availability section and make sure that all deposited datasets are public latest upon online publication of the manuscript.
- Please check again that the number "n" for how many independent experiments were performed, their nature (biological versus technical replicates), the bars and error bars (e.g. SEM, SD) and the test used to calculate p-values is indicated in the respective figure legends. Please also check that all the p-values are explained in the legend, and that these fit to those shown in the figure. Please provide statistical testing where applicable. Please avoid the phrase 'independent experiment' but clearly state if these were biological or technical replicates. Please also indicate (e.g. with n.s.) if testing was performed, but the differences are not significant. In case $n=2$, please show the data as separate datapoints without error bars and statistics. If $n<5$, please show single datapoints for diagrams.

Presently, it seems that no statistical testing was performed for any of the diagrams (with $n>2$) shown in the main, EV or Appendix figures. Please do that and also add a paragraph to the methods section explaining the statistical methods used. See also:

<http://www.embopress.org/page/journal/14693178/authorguide#statisticalanalysis>

- Please add to each legend (main, EV and Appendix figures, where applicable) a 'Data Information' section (or name the provided section like this) explaining the statistics used or providing information regarding replicates and scales. See:

- Please add the information provided in the Appendix Tables S1, S2 and S3 directly to the Reagents and Tools Table. Then please remove the tables from the submission system and also remove their legends from the Appendix file. Please also add callouts to the table where appropriate.
- Please add Appendix Table S4 into the Appendix file next to its legend. Do not upload this separately.
- Appendix Tables S5, S6 and S7 are datasets. Please name and upload these as Dataset EV5, Dataset EV6 and Dataset EV7 in all places (source file names, titles in the submission system and their callouts). Please remove their legends from the Appendix. Instead, please put the respective legend on the first TAB of the corresponding Excel file.

In addition, I would need from you uploaded separately:

Best,

Referee #1:

I am statisfied with the revised manuscript. The authors answered all my questions/concerns.

Referee #2:

Authors have carefully considered the reviews. In my opinion the manuscript is now ready for publication.

Biochemistry
UNIVERSITY OF TORONTO

July 16, 2025

Dear Dr. Breiling,

Thanks again for the timely and productive review of, “**A bacterial toxin-antitoxin system involved in an unusual response to genotoxic stress.**” We thank you and the reviewers for the constructive feedback and suggestions.

We have completed the requested revisions and now return to you our revised manuscript.

One point that we would like to note is regarding the request to show raw data for all plots where $n=2$. In our previous revised manuscript, we changed all main figure panels with $n=2$ to show the raw data (as this request was made by one of our reviewers) except for panels 4F and 7B. This was because neither plot is conducive to showing non-averaged data, as 4F is a heatmap and 7B has a large number of data points even with averaging. Therefore, we chose to present these data as averages to maximize clarity for the readers of the paper, and additionally because the raw data are provided in the source data files.

There are several panels in the Extended View figures that are also plotted from $n=2$ data and we have again opted to keep these figures with averaged data. This is because many of these plots are composed of dozens of data points even after averaging, and consequently displaying the raw data would either render them unreadable or lead to individual data points being obscured due to the density of points being plotted. Our concern is that in this instance it would risk making the data we are presenting uninterpretable to readers. Please advise us as to whether this is acceptable as including each datapoint for these Extended View figures may require us to explore alternative ways of presenting the data.

We thank you again for your continued support of our manuscript and are happy to discuss the revisions further if you should have any concerns or additional requests.

Warm regards,

Alexander W. Ensminger
alex.ensminger@utoronto.ca
416-978-6522

Jordan D. Lin
jdolin@stanford.edu

Prof. Alexander Ensminger
University of Toronto
Biochemistry, Molecular Genetics
661 University Ave
Toronto M5G 1M1
Canada

Dear Prof. Ensminger,

I am very pleased to accept your manuscript for publication in the next available issue of EMBO reports. Thank you for your contribution to our journal.

Yours sincerely,
